

# A high-resolution dataset of Rock Glaciers in the Peruvian Andes (PRoGI): inventory, characterization and topoclimatic attributes.

Katy Medina[1,2,6], Hairo León[1,2], Edwin Badillo-Rivera[1,3,4], Edwin Loarte[1,2], Xavier Bodín[5] and José Úbeda[6,7]

[1]Research Center for Environmental Earth Science and Technology (ESAT), Santiago Antunez de Mayolo National University (UNASAM), Huaraz, 02002, Peru
[2]Faculty of Environmental Sciences, Santiago Antunez de Mayolo National University (UNASAM), Huaraz, 02002, Peru
[3]Faculty of Environmental Engineering and Natural Resources, National University of Callao, Bellavista, 07011, Peru
[4]Research Center Climate Change and Disaster Risk Management, National University of Callao, Bellavista, 07011, Peru
[5]Laboratoire EDYTEM, Université Savoie Mont Blanc, CNRS, Le Bourget-du-Lac, 73370, France.
[6]Departamento de Geografía. Universidad Complutense de Madrid   28040 Madrid, Spain.
[7]Departamento de Ciencias de la Tierra, Guías de Espeleología y Montaña. Casilla del Mortero. 28189 Torremocha de Jarama, Spain.

*Correspondence to*: Katy Medina (kmedinam@unasam.edu.pe)

**Abstract.** Rock glaciers are key periglacial landforms in high mountain systems, serving as indicators of permafrost, contributors to mountain hydrology, and sentinels of climate change. Despite their scientific and practical importance, detailed knowledge of their distribution, characteristics, and dynamics in the Peruvian Andes remains limited. This study presents the Peruvian Rock Glacier Inventory (PRoGI v1.0), – a comprehensive, high-resolution inventory of rock glaciers covering the

entire Peruvian Andes, encompassing their spatial distribution, morphological attributes, and topoclimatic controls. Unlike previous local-scale studies, PRoGI v1.0 provides national-scale coverage using standardized methods aligned with International Permafrost Association (IPA) guidelines and updated data. Using sub-meter satellite imagery (Bing Maps 2024 and Google Earth 2017) and IPA classification standards, we mapped 2338 rock glaciers with a total area of 94.09 ± 0.05 km². Approximately 31 % of these landforms were classified as active, 49 % as transitional, and 20 % as relict. They predominantly

occur between ~4416 and 5783 m a.s.l. (mean elevation ~4999 m) on slopes averaging ~20.7° (range 7–37°). Spatially, rock glaciers are concentrated in the southern Peruvian Andes, with sparse distribution in central and northern Peru. Most have a southern to southwestern aspect (predominantly S, SW, and SE-facing), and the lower limit of permafrost (indicated by the lowest active rock glacier fronts) is ~3541 m a.s.l. Our inventory serves as a benchmark dataset that significantly advances the understanding and monitoring of mountain permafrost, and it provides a basis for assessing the hydrological importance of

rock glaciers in the Peruvian Andes under climate change scenarios. The dataset is available at https://doi.pangaea.de/10.1594/PANGAEA.983251 (Medina et al., 2025a).



# 1 Introduction

The periglacial environment, characterized by cold, non-glacial conditions, represents one of Earth's most dynamic and
climate-sensitive landscapes. Dominated by freeze-thaw cycles, permafrost dynamics, and distinctive landforms such as
patterned ground, solifluction lobes, and rock glaciers (French, 2017), these environments play a critical role in high-mountain
hydrology, biodiversity, and geomorphology. Recent studies highlight their exceptional vulnerability to climate change, with
rising temperatures driving permafrost degradation and destabilizing landforms across global mountain systems (Haeberli et
al., 2017; IPCC, 2022). Nowhere is this more evident than in the tropical Andes, where elevations above 5000 m a.s.l. have
warmed at ~0.17 °C per decade since the mid-20th century (Aguilar-Lome et al., 2019), threatening water security and
ecosystem stability.

Among periglacial landforms, rock glaciers stand out as both geomorphological archives and vital water reservoirs. These ice-
debris landforms, formed by the creep of ice-rich permafrost, exhibit diagnostic steep fronts, lateral margins, and ridge-and-
furrow surface topography (RGIK, 2023). Comprising 15–70 % ice by volume (Halla et al., 2021; Haq and Baral, 2019), rock
glaciers store substantial water equivalents in arid regions like the southern Peruvian Andes (Boccali et al., 2019; Janke et al.,
2017; Rangecroft et al., 2015). Their debris mantle confers thermal inertia, buffering ground ice against short-term climate
variability (Brighenti et al., 2021). This dual role as climate sentinels and hydrological buffers makes rock glaciers
indispensable for understanding long-term environmental change. Mountain permafrost, defined as ground remaining ≤0 °C
for at least two consecutive years, underpins these systems. It stabilizes steep slopes, modulates groundwater flow, and sustains
ecosystems (Gruber and Haeberli, 2007). However, mountain permafrost is highly sensitive to warming; rising temperatures
lead to permafrost degradation and can trigger the release of stored greenhouse gases (Biskaborn et al., 2019). In the Andes,
where glacial retreat has increased the relative importance of permafrost as a water resource, its hydrological role remains
critical yet poorly quantified due to sparse observations in remote high-altitude areas.

Rock glaciers serve as direct visual indicators of mountain permafrost, with their presence delineating the occurrence of ground
ice and the approximate lower limits of discontinuous permafrost (Brenning, 2005). Along the higher South American Andes
(>4000 m a.s.l.), studies in Argentina, Chile, and Bolivia have leveraged rock glacier inventories to map permafrost and assess
water storage (Azócar and Brenning, 2010; Esper Angillieri, 2017; Falaschi et al., 2015; Rangecroft et al., 2015). However
knowledge gaps still persist, in Peru: existing inventories are fragmented (Badillo-Rivera et al., 2021; León et al., 2021) and
lacking standardized methods or detailed topoclimatic analyses. To address this, we present the Peruvian Rock Glacier
Inventory (PRoGI v1.0), the first high-resolution, nationally comprehensive rock glacier dataset for the Peruvian Andes,
compiled using the mapping standards of the International Permafrost Association's Action Group (IPA) on Rock Glacier
Inventories and Kinematics (RGIK, 2023). By combining sub-meter remote sensing imagery with rigorous geospatial analysis,
PRoGI v1.0 documents the distribution, morphology and climatic controls of rock glaciers across Peru.

This new dataset enables advances in permafrost modelling and can be integrated into water resources assessments and disaster
risk management efforts in the Peruvian Andes by environmental authorities (e.g. National Water Authority (ANA), National



Center for Estimation, Prevention, and Disaster Risk Reduction (CENEPRED). In addition, by adhering to standardized IPA criteria, the inventory is interoperable with other international databases, representing a significant step toward a global rock glacier database." Splitting it up would reduce run-on complexity.

The following sections describe the study area and data sources (Sect. 2–3), the methodology for inventory compilation (Sect. 4), the resulting inventory characteristics (Sect. 5), and their broader implications (Sect. 6), followed by conclusions.

## 2 Study area

The Peruvian Andes (spanning ~4° to 18°S and 80º to ~69°W ) are the central sector of the tropical Andes, a mountain range that plays a critical role in regional hydrology and climate regulation (Vuille et al., 2018). This region is characterized by dramatic topography, with elevations reaching 6757 m a.s.l. on Nevado Huascarán, over an area of approximately 308 124 km² (about 30 % of Peru's territory; INAIGEM, 2023). The climate of the Peruvian Andes is highly heterogeneous shaped by global atmospheric circulation and local topography. The eastern (Amazon-facing) slopes of the Andes receive moist air masses from the Amazon Basin, resulting in steep precipitation gradients (up to ~1100 mm/year in windward areas) and generally humid conditions. In contrast, the western slopes are much drier and colder, often receiving less than ~500 mm/year, due to the rain-shadow effect of the Andes (Garreaud, 2009). These sharp climatic gradients, combined with the extreme elevation range, create a mosaic of microclimates that influence the distribution and characteristics of periglacial landforms (including rock glaciers).

Bonshoms et al. (2020) divided the Peruvian Andes into four climatological subregions (Fig. 1) based on precipitation and temperature regimes. We adopt these four subregions as the basis for our analysis, identifying them as the Northern Wet Outer Tropics (NWOT), Northern Dry Outer Tropics (NDOT), Southern Wet Outer Tropics (SWOT) and Southern Dry Outer Tropics (SDOT). Each subregion encompasses the mountains covered by current glaciers   as well as surrounding highland areas bellow the glaciers. These subregional divisions facilitate comparisons of rock glacier characteristics under different climate settings across Peru.

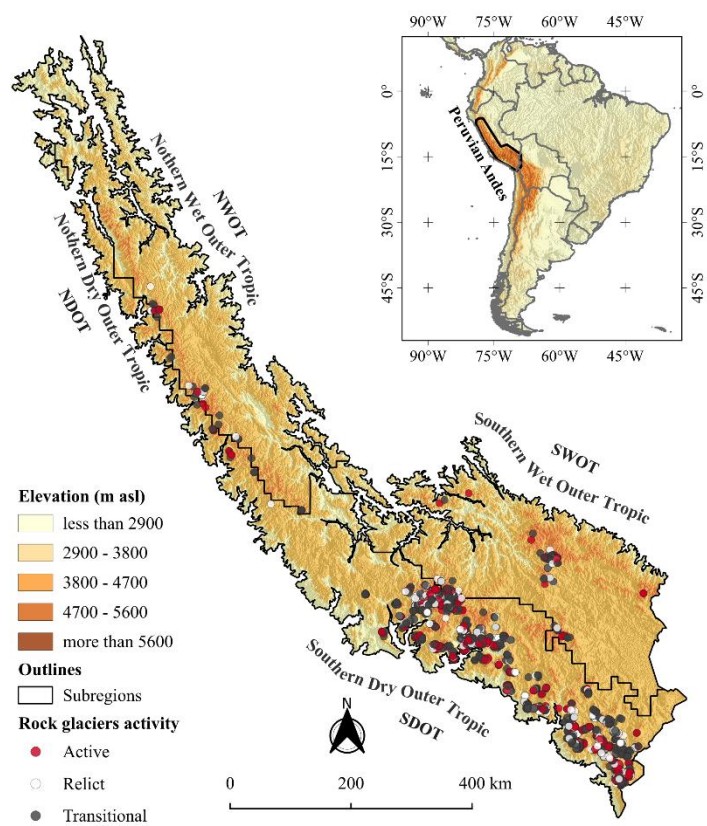

**Figure 1.** Distribution of active, transitional and relict rock glaciers in the Peruvian Andes overlaid on a 30 m ALOS DEM. The inset map highlights the location of the study area in South America.

# 3 Data

## 3.1 Data sources

The rock glacier inventory was manually compiled using high-resolution satellite imagery. Bing Maps Aerial imagery was used as the primary data source for mapping, and Google Satellite imagery was used in areas where Bing imagery was unavailable or of worst quality. We accessed these images within a GIS environment (using the *QuickMapServices plugin* in QGIS v3.40) to facilitate consistent mapping and geodatabase creation (Rangecroft et al., 2014). The imagery has high spatial resolution (Table 1), allowing detailed identification of rock glacier features. In a few cases, we also consulted historical Google Earth images (from earlier dates or different seasons) to identify rock glaciers in locations where seasonal snow cover, cloud or shadows hid the landforms in the primary imagery (Pandey, 2019). Table 1 summarizes the main imagery sources and acquisition dates used for rock glacier mapping. In total, 2338 rock glaciers were mapped using these optical datasets



(2095 from Bing and 243 from Google imagery), which provided nearly complete high-resolution coverage of the Andes of Peru.

**Table 1.** Image sources and acquisition dates.

| Image source | Sensors | Spatial resolution | Acquisition date | Rock glacier mapped |
|---|---|---|---|---|
| Bing Satellite | Maxar Technologies | ~ 1 to 3 m | November 2024 | 2095 |
| Google Satellite | CNES/Airbus | ~ 1 to 5 m | April 2017 | 243 |

### 3.2 Topoclimatic datasets

To analyse the topographic and climatic context of rock glacier distribution, we compiled several auxiliary datasets:

(1) Digital Elevation Model (DEM): we used the 12.5 m-resolution ALOS PALSAR for Peru from Japan Aerospace Exploration Agency DEM (Rosenqvist et al., 2007) to derive topographic attributes (elevation, slope, aspect) for each rock glacier.

(2) Climate data (temperature and precipitation): Mean Annual Air Temperature (MAAT) and Annual Precipitation (AP) 110 were obtained from the CHELSA 2.1 climate dataset at 0.1° (~ 10 km) resolution (Karger et al., 2017). We extracted the 1979-2013 climatology (CHELSA provides high-resolution modelled climate surfaces). The CHELSA data have been widely used for characterizing rock glaciers climates in regional studies and show good correlation with ground meteorological observations (Anderson-Teixeira et al., 2015). These temperature and precipitation values allowed us to approximate the climatic regime (e.g., cold-dry vs warm-wet) in each rock glacier´s location (Drewes et al., 2018; Esper 115 Angillieri, 2017; Haq and Baral, 2019; Rangecroft et al., 2014).

(3) Permafrost model data: we included Mean Annual Ground Temperature (MAGT) data for the period 2000-2016 from Obu et al. (2019). This modelled permafrost data product helped indicate which rock glaciers are located within zones of likely permafrost (MAGT $\leq$ 0 °C) and provided an independent check on our activity classifications.

All spatial datasets were co-registered and clipped to the extent of the Peruvian Andes. We used these layers to extract 120 topoclimatic attributes (elevation, slope, aspect, MAAT, AP, MAGT) for each rock glacier, which are analysed in Section 5.3.

### 3.3 Auxiliary data

To distinguish rock glaciers from other ice-related landforms, we also utilized existing inventories of glaciers in Peru. In particular, we employed data from the National Inventory of Glaciers and Glacial Lakes (INGLOG II) of Peru (INAIGEM, 125 2023), which provides outlines of both clean glaciers and debris-covered glaciers. By overlapping the glacier outlines from INGLOG II on the rock glaciers polygons, it allows avoid mistaking debris-covered glaciers for rock glaciers during mapping. This additional reference was especially useful in complex high-mountain environments where rock glaciers are often on the

same slope as present-day glaciers. In the database, the cases where a rock glacier is immediately downstream of a glacier was flagged. to ensure no double-counting or misclassification occurred.

## 4 Methodology

### 4.1 Identification and mapping of rock glaciers

The mapping approach followed the official guidelines of the International Permafrost Association (IPA) Action Group on Rock Glacier Inventories and Kinematics (RGIK) for inventorying rock glaciers (RGIK, 2023). To ensure thorough coverage and consistency, we implemented a systematic mapping workflow following the next steps (4.1.1.-4.1.3.):

### 4.1.1 Grid-based systematic mapping

The entire study region was divided into a grid of 50 × 50 km cells to cover the Peruvian Andes uniformly. Each grid cell was examined in detail using the high-resolution satellite imagery described above. We primarily used Bing Satellite imagery within QGIS 3.40 (https://qgis.org/project/overview/, accessed on 1 April 2025) for visual scanning of each cell (Jones et al., 2018, 2021). If a given area in Bing had cloud cover, snow obstruction, or otherwise poor image quality, we switched to Google Earth imagery for that grid cell. This grid-based approach ensured that no areas were overlooked, and it helped organize the work among the mapping team.

### 4.1.2 Digitization protocol

When a rock glacier was identified in the imagery, we delineated it as a polygon following a consistent digitization criteria. Each rock glacier was assigned a unique identifier derived from its geographic coordinates (specifically, an ID based on the centroid latitude and longitude, e.g., "RGU154384S0729196W" as illustrated in the data dictionary). We then manually digitized a polygon to capture the entire rock glacier landform. The mapping of each feature included:

(1) Upslope extent: the rooting zone or talus contributing area, up to the point where distinct break in slope or change in surface texture marked the upper boundary of the rock glacier

(2) Main body: the central part of the rock glacier characterized by its typical surface morphology (e.g., longitudinal ridges and furrows, or a lobate debris structure).

(3)  Frontal and lateral margins: the steep front (toe) and sides of the rock glacier, which often have sharper convex profiles or abrupt edges separating the landform from surrounding terrain.

In some cases, rock glaciers exhibited very extended or degraded fronts (e.g., a "tongue-shaped" that had flowed out and thinned, or a collapsed snout). In such situations, we adopted a conservative mapping approach to remain consistent with IPA guidelines for degraded features (RGIK, 2023).

The landform discernible boundary was drawn (e.g., along collapse features or the farthest ridge) and the polygon was delimited so that it did not extend more than 50 m beyond that line. This rule prevented overestimation of area for partially degraded rock glaciers. Fig. 2 illustrates an example of how an extended/collapsed front was constrained. After delineation,

the date and source of the imagery used for each polygon was recorded, to document the temporal coverage of the inventory. Table 3 includes an "Imagery Date/Source" attribute for each entry.

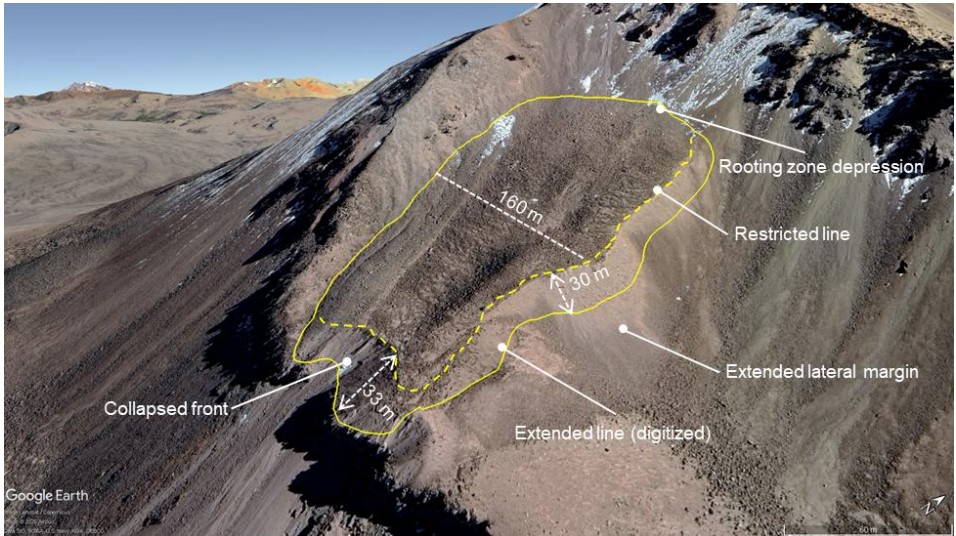

**Figure 2**. Extended/collapsed front mapping example. on: Location of the rock glacier shown 17°04'30″ S, 69°59'24″ W. Basemap: © Google Earth.

### 165    4.1.3 Quality Control

After initial mapping, we undertook a quality control process to improve the accuracy and completeness of the inventory. First, we assigned a certainty level to each mapped polygon: "1" for high-confidence rock glaciers and "0" for features where identification was uncertain. Approximately 12 % of the polygons (281 features) were flagged as low certainty due to factors such as snow cover, shadows, or borderline morphology, that made interpretation difficult. All low-certainty cases were

subjected to careful review and, if necessary, remedial mapping actions. To be applied during this quality control stage, four possible review operations were defined, similar to the procedure described by Sun et al. (2024).

- (Sun et al., 2024)Remain: no change needed – the polygon accurately delineates a rock glacier and is confirmed as correct.
- Remove: delete polygon if it was determined not to be a rock glacier (e.g., a misidentified landform such as a solifluction lobe or a pronival rampart). These misidentified features are usually associated with the downslope margins of perennial

or semi-permanent snow patches (Matthews et al., 2011) and can exhibit superficial ridge-like forms due to processes like snow push, solifluction, or rockfalls from nearby scarps, which slide over frozen snow and accumulate at the base of the slab. However, they lack the internal ice thickness and other diagnostic traits of rock glaciers (Colucci et al., 2016). We



also cross-checked the national glacier inventory (INGLOG II) to ensure that any feature in question was not actually a debris-covered glacier.

- Refine: modify the polygon if the feature is indeed a rock glacier but our original delineation missed part of it or included extraneous parts. This could involve adjusting boundaries to more closely match the visible geomorphological edges.
- Retrieve: add a new polygon for any previously overlooked rock glacier. During the review, some subtle or small rock glaciers that had been missed in the first pass was discovered, particularly in areas of complex terrain or where imagery was less clear. In total, 32 rock glaciers were "retrieved" in this manner. Additionally, if a single mapped polygon was
found to actually contain multiple adjacent rock glaciers (i.e., a compound feature), it was split into separate units when distinct lateral boundaries could be discerned in the imagery.

Each low-certainty or complex case was re-examined by the team, and the appropriate operation (remain, remove, refine, or retrieve) was applied. After this systematic check, a cleaned and verified set of rock glacier polygons was achieved. The quality control review significantly improved the inventory's reliability, reducing spurious inclusions and omissions.

**4.2 Geomorphological identification criteria**

The identification of rock glaciers was based on standard geomorphological criteria defined by the IPA guidelines (RGIK, 2023). Diagnostic surface features and morphology were used to classify each landform's activity state (active, transitional, relict) during mapping. In summary, the key geomorphic indicators included:

- Surface flow structures: well-developed ridges and furrows on the rock glacier surface indicate past or present creep
movement. Active rock glaciers typically show sharp, well-defined ridge–furrow systems, whereas transitional ones have more subdued topography, and relict rock glaciers have only faint or no flow structures remaining.
- Terminus slope shape: steep, over-steepened snouts (often ~30–40°) with an abrupt break to the upper surface are characteristic of active rock glaciers, due to ongoing internal ice support. Transitional rock glaciers might have slightly gentler fronts or minor vegetative cover, and relict rock glaciers usually display lower-angle toes with significant
smoothing and sometimes vegetation growth.
- Rock glacier body morphology: a swollen or bulging body cross-section suggests internal ice presence; some active rock glaciers even exhibit exposed ice or thermokarst depressions. Transitional forms may still appear bulges but with fewer signs of ice, and relict forms often have a flattened, collapsed appearance with possible surface subsidence features,indicative of thickness loss or disappearance of ice.
- Surface color and weathering: Active rock glaciers often have a relatively fresh appearance – for example, a cleaner-looking frontal slope with less weathered rocks, and a darker, dustier upper surface. Relict rock glaciers tend to have more uniformly weathered surfaces (darker "varnished" appearance) and might even develop edaphic soil and lichen cover. The transitional forms, on the other hand, show some areas of the front with fresh material, but at the same time darker fronts with signs of weathering.



▪    Vegetation presence: active rock glaciers are typically free of vegetation on their surface due to constant movement and ice presence. Transitional rock glaciers might support some pioneer vegetation around their margins or in stable sections, while relict rock glaciers can have significant vegetation (grasses, shrubs) on the surface and around, reflecting long-term stability.

These criteria (summarized in Table 2) guided the visual interpretation of each landform and were consistently applied to
ensure the classification of activity state and morphology in the inventory was as objective as possible. Any marginal cases were noted for further scrutiny (often with assistance from the external reviewers, as described below).

**Table 2.** Geomorphological indicators for activity status.

| Geomorphological indicator | Active | Transitional | Relict |
|---|---|---|---|
| Surface flow structure | Defined furrow and ridge topography | Relatively subdued micro-topography | Less defined furrow and ridge topography |
| Front slope | Steep (>30 to 35°) Abrupt transition (sharp-crested) to the upper surface | Moderately steep fronts (generally gentler than active) | Gently sloping (<30°) Gentle transition) to the upper surface |
| Rock glacier body | Swollen body Surface ice exposures | Can have swollen body and surface ice exposures | Flattened body Surface collapse features |
| Surface colour | Light-colored (little clast weathering) frontal zone, and a darker varnished upper surface | Dark-coloured rock-varnished frontal slopes | Front has a well-defined color with signs of erosion |
| Vegetation presence | No vegetation on the surface | There is no vegetation on the surface, but it may exist in the surrounding area | Presence of vegetation on the surface and in the surrounding area |

### 4.3 Inventory compilation and validation

The mapping and review process was carried out by a team consisting of five primary cartographers (the core authors of this
study) and sixteen independent expert reviewers with experience in rock glacier inventories. The inventory was divided into regional sub-areas, and one or more reviewers were assigned to each sub-area for an external check. A complete list of expert reviewers and their affiliations is provided in Table S1 (Supplement). The reviewers examined the draft inventory and provided feedback, marking any polygons that might require one of the four quality-control operations described above (remain, remove, refine, retrieve).

Each suggested change was discussed and then implemented by the mapping team, ensuring that the final decisions were internally consistent across the entire inventory. When the characteristics of a particular rock glacier were uncertain or ambiguous in Bing imagery, we used Google Earth's high-resolution imagery as a supplementary perspective to confirm features like flow structures or boundaries. After incorporating all reviewer comments and finalizing the polygon delineations,



we compiled the attribute table for the inventory. For each rock glacier unit, we recorded a comprehensive set of attributes (see

Table 3 for full definitions). Key attributes include:

- Location information: Internal ID, coordinates (latitude/longitude), and the corresponding region and subregion of the Andes (NWOT, NDOT, SWOT, SDOT).

- Hydrological context: The name of the major river basin and local watershed in which the rock glacier is located.

- Activity state: Categorical attribute (active, transitional, relict) along with the criteria/evidence used for this assessment

(e.g., presence of flow structures, slope thresholds, etc.).

- Morphology: Shape classification (tongue-shaped, lobate, coalescent, polymorphic) and genesis (origin as glaciogenic or cryogenic, if determined).

- Dimensions: Planimetric area of the rock glacier (km²) and elevations (minimum, maximum, mean elevation) derived from the DEM.

- Slope: Mean slope angle of the rock glacier surface (in sexagesimal degrees).

- Mapping details: Data source and date of imagery used for mapping, the mapper's initials, and the reviewer's initials, as well as a "certainty" flag (as described above) and any additional notes.

All attributes were recorded in a GIS shapefile and cross-checked for consistency. The attribute schema is outlined in Table 3 for reference. This rich database structure ensures that PRoGI v1.0 is not just a collection of polygons, but a data-rich inventory

that can be queried for spatial patterns, compared with other inventories, and used for quantitative analyses of Peru's rock glaciers.

**Table 3.** Detailed attribute table of polygons representing rock glaciers. M: mandatory attribute and O: optional attribute.

| Attribute | Description | Type [Values] | Source |
|---|---|---|---|
| fid (M) | Internal ID:<br>Unique identifier of each rock glacier. | Automatic filling | |
| Primary ID (M) | Rock glacier ID:<br>RGU + 12 to 15 digits depending on the "Lat", "Lon" values. Always 4 decimal places after the degrees.<br>(e.g., RGU153844S0729196W means 15.3844° South and 72.9196° West) | Automatic filling | RGIK (2023): section 5.2 |
| Sour_Data (M) | Source data:<br>Source data used to digitize rock glacier outlines<br>                              Bing Satellite | Text [Bing Satellite, Google Satellite] | |
| Map_Date (M) | Date of mapping:<br>Date on which the rocky glacier was digitized<br>(Format: YYYY-MM-DD UTC-5) | Date | RGIK (2023): section 5.3 |
| Mapper (M) | Mapper's name:<br>Name of the operators who performed the digitization of rock glaciers. | Text | |





| | | | |
|---|---|---|---|
| Reviewer (M) | Reviewer's name:<br>Name of the reviewers who verified the rock glacier outlines. | Text | |
| Addi_Inf (M) | Additional information:<br>This attribute allows recording some elements of the environment that may be of interest for future studies (e.g., proximity to wetlands, roads, etc.) | Text | |
| Region (O) | Region of rock glacier:<br>Peru is divided into 24 political regions, of which the rock glacier inventory is distributed in 9. | Text | (IGN, 2023) |
| Subregion (O) | Subregion of rock glacier:<br>Name of the subregion where the inventoried rock glaciers are located. Based on Peru's temperature and precipitation patterns. | Text [<br>1. Southern Dry Outer Tropic.<br>2. Southern Wet Outer Tropic.<br>3. Northern Dry Outer Tropic.<br>4. Northern Wet Outer Tropic.] | Bonshoms et al. (2020) |
| Hydro_Wat (O) | Hydrographic watershed of rock glacier:<br>Name of the level 1 hydrological watershed (Pfafstetter coding) where the inventoried rock glaciers are located. | Text [<br>1. Región Hidrografica del Pacífico.<br>2. Región Hidrografica del Amazonas.<br>3. Región Hidrografica del Titicaca.] | (ANA, 2003) |
| River_Basi (O) | River basin of rock glacier:<br>Name of the level 4, 5 or 6 river basins (Pfafstetter coding) where the inventoried rock glaciers are located. | Text | (ANA, 2003) |
| Activity (O) | Activity of rock glacier:<br>Activity class assigned to the rock glacier. | Text [<br>1. Active.<br>2. Transitional.<br>3. Relict.] | RGIK (2023): sections 3.4, 5.3 and 6.1 |
| Act_Assess (O) | Criteria for activity assessment:<br>Criteria used for the evaluation of activity classes. | Text | RGIK (2023): section 3.4.2 |
| Geometry (O) | Morphology of rock glacier:<br>Defines the morphology of the identified rock glacier. | Text [<br>1. Tongue-shaped.<br>2. Lobate.<br>3. Coalescent.<br>4. Polymorphic.] | Humlum (1982); Krainer and Ribis (2012) |
| Delin_type (O) | Delineation type of rock glacier:<br>Extended geomorphological footprint that includes the outer parts (frontal and lateral margins). | Text | RGIK (2023): sections 3.6 and 5.4 |
| Lon (M) | Longitude of the rock glacier centroid in decimal degrees. | Automatic filling<br>[-77° to -69°] | |



| Lat (M) | Latitude of the rock glacier centroid in decimal degrees. | Automatic filling [-18° to -10°] |
|---|---|---|
| Area (O) | Rock glacier area (km$^2$): Values obtained by field calculator tool in QGIS 3.40. | Automatic filling [0.001 to 0.89] |
| Elev_mean (O) | Mean elevation of rock glaciers (m a.s.l.): Values extracted by zonal statistics (mean) in QGIS 3.40 using the ALOS PALSAR DEM (12.5 m). | Automatic filling [4416 to 5783] |
| Elev_min (O) | Minimum elevation of rock glaciers (m a.s.l.): Values extracted by zonal statistics (minimum) in QGIS 3.40 using the ALOS PALSAR DEM (12.5 m). | Automatic filling [3541 to 5657] |
| Elev_max (O) | Maximum elevation of rock glaciers (m a.s.l.): Values extracted by zonal statistics (maximum) in QGIS 3.40 using the ALOS PALSAR DEM (12.5 m). | Automatic filling [4477 to 5977] |
| Slope (O) | Mean slope of rock glacier (Degrees): Values obtained using the slope tool and zonal statistics (mean) in QGIS 3.40 using the DEM. | Automatic filling [7° to 37°] |
| Aspect (O) | Aspect of rock glacier (Degrees): Values obtained using the aspect tool and zonal statistics (mean) in QGIS 3.40 using the DEM. | Automatic filling [22° to 335°] |
| MAAT (O) | Mean Annual Air Temperature (°C): Values extracted by zonal statistics (mean) in QGIS 3.40 using the bio12 variable from CHELSA 2.1 (1 km). | Automatic filling [-3.0 to 5.3] |
| MAGT (O) | Mean Annual Ground Temperature (°C): Values extracted by zonal statistics (mean) in QGIS 3.40 using the ground temperature raster (1 km) obtained by Obu et al. (2019). | Automatic filling [-0.1 to 10.9] |
| AP (O) | Annual precipitation (mm): Values extracted by zonal statistics (sum) in QGIS 3.40 using the bio1 variable from CHELSA 2.1 (1 km). | Automatic filling [315 to 2389] |
| PISR (O) | Annual Potential Incoming Solar Radiation (kWh m$^{-2}$): Values obtained using the Potential Incoming Solar Radiation tool of SAGA GIS 9.1 from DEM and zonal statistics (mean) in QGIS 3.40. | Automatic filling [1744 to 2727] |

## 4.4 Classification of rock glaciers

When classifying rock glaciers by activity state, the morphological scheme proposed by Barsch (1996) was used, in combination with guidelines from the IPA Action Group on Rock Glacier Inventories and Kinematics (RGIK, 2023). As a result, three categories of rock glacier activity were distinguished: active, transitional, and relict (Fig. 3). In the absence of



kinematic data, active rock glaciers are defined as landforms containing interstitial ice (Roer et al., 2005; Wirz et al., 2016). Consequently, they tend to have a topography dominated by pronounced ridges and furrows created by longitudinal push and transverse compression (Charbonneau and Smith, 2018; Sattler, 2016), exhibit a frontal slope steeper than about 30–35°, and usually lack vegetation (Tielidze et al., 2023). Transitional rock glaciers (also referred to as inactive) may still contain ice but have ceased moving. They typically have a gentler frontal slope (< 30–35°) and display geomorphological indicators of inactivity, such as a more subdued microtopography and smoother, dark-colored, rock-varnished frontal and lateral margins (Ahumada et al., 2014). In addition, transitional rock glaciers often support more vegetation on their surface, terminus, or surroundings (Brenning, 2005). They may be climatically inactive (transitional due to ice melt) or dynamically inactive due to reduced input of talus and/or ice (nourishment) to the system (Kellerer-Pirklbauer et al., 2012). Relict rock glaciers, which show no evidence of recent movement, are characterized by collapse structures on their surfaces due to the melting of remaining ice content (Abdullah and Romshoo, 2024). They also exhibit subtle micro-relief and shallow, round-crested frontal and lateral slopes (RGIK, 2023; Scotti et al., 2013). Typically, relict rock glaciers have concave longitudinal profiles (Colucci et al., 2016) and compared to active and transitional forms, they often have vegetated surfaces and are usually found at lower elevations (Baroni et al., 2004; Scotti et al., 2013).

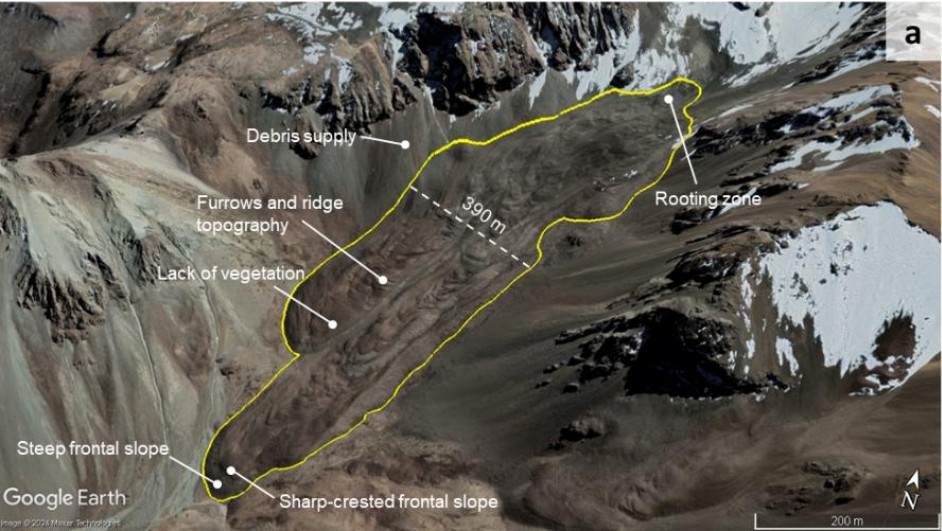



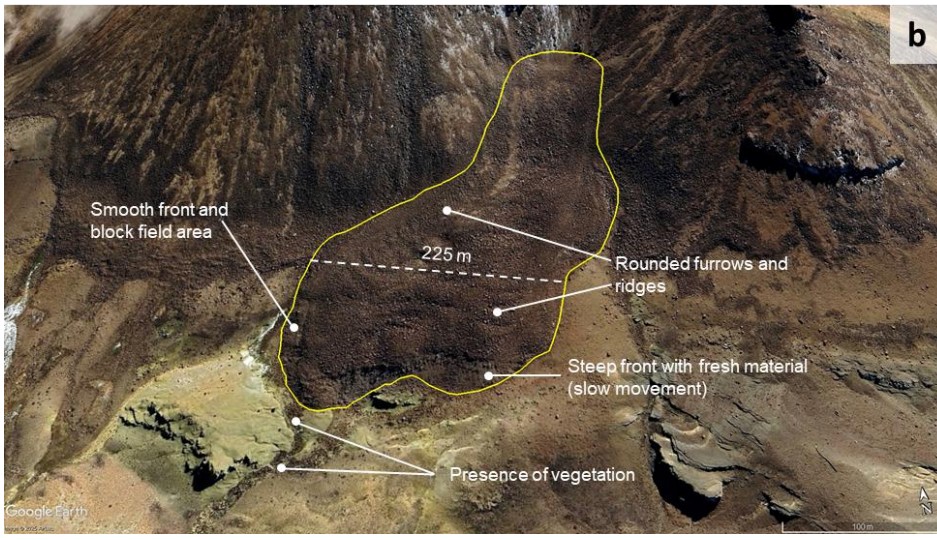

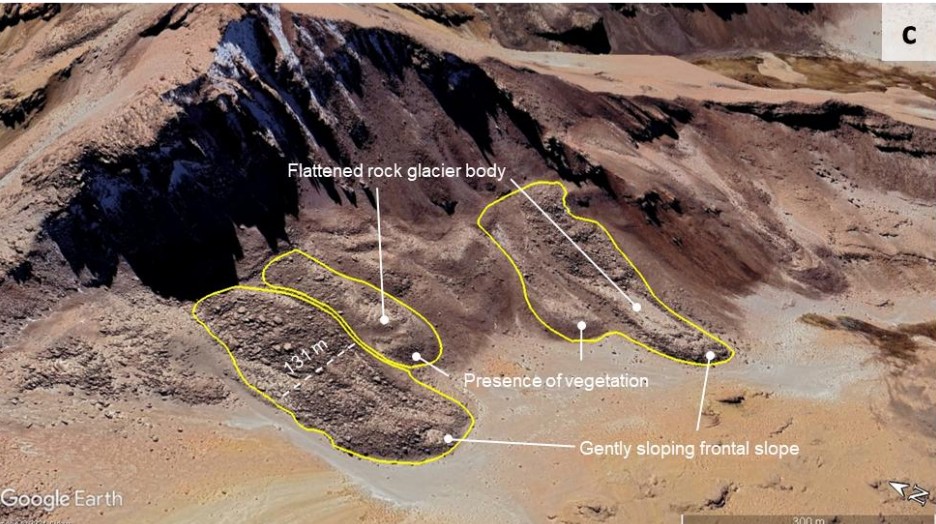

**Figure 3.** Rock glacier mapping and classification examples: (a) active rock glaciers (15°25'31.85"S, 72°10'31.68"W); (b) transitional rock glacier (17° 7'58.62" S, 69°52'24.82" W); and (c) relict rock glacier (16°29'6.81"S, 71° 8'36.74"W). Yellow outline indicates the rock glacier extent and dashes white line illustrates the width. Basemap: © Google Earth.

In addition to activity, we classified rock glaciers by their planform geometry as tongue-shaped, lobate, coalescent, or polymorphic. A length-to-width ratio greater than 1 indicates a tongue-shaped rock glacier, whereas a ratio less than 1 indicates a lobate rock glacier (Harrison et al., 2008; Humlum, 1982). Coalescent rock glaciers are composite features by the convergence of several tongue-shaped lobes, typically exhibiting a spatulate (flared) terminus (Humlum, 1982). In contrast, polymorphic rock glaciers refer to assemblages with heterogeneous morphological characteristics, often displaying multiple planimetric shapes within a single system (Falaschi et al., 2015). The specific criteria used to classify rock glaciers by



geometry, along with illustrative examples, are provided in the Table 4. The images shown were taken from Bing Maps

imagery (© Microsoft Corporation).

**Table 4.** Criteria for the classification of rock glaciers based on their geometry.

| Geometry class | Criteria | Example |
|---|---|---|
| Tongue-shaped | Relationship length/width > 1 | Lat/Lon: 15°25'34.11"S 72°11'31.08"W |
| Lobate | Relationship length/width < 1 | Lat/Lon: 12°35'15.39"S, 75°48'6.38"W |
| Coalescent | They are various tongue-shaped landforms with a spatulate flaring at the end and do not have well-defined margins to be individualized. | Lat/Lon: 16°10'12.88"S, 70°57'14.45"W |



| Polymorphic | Based on their composition, here are landforms in which the length/width ratio is greater and less than 1 (tongue-shaped and lobate) | 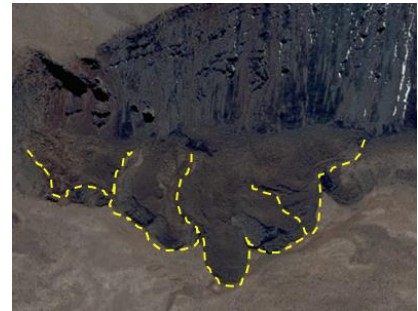 |
|---|---|---|

Lat/Lon: 17°24'32.35"S, 69°40'57.41"W

## 4.5. Topoclimatic features

For each rock glacier, a suite of topographic characteristics and climatic variables was extracted. Topographic attributes include geographic coordinates (latitude and longitude), minimum, mean and maximum elevations, and mean slope and aspect in the rock glacier centroid. These parameters were derived tn QGIS by superimposing the mapped rock glacier polygons on a digital elevation model (DEM) and applying zonal statistics for each polygon. The minimum elevation corresponds to the lowest point within a rock glacier's outline (generally at the front), while the maximum elevation is the highest point (typically at the rooting zone). The mean elevation was calculated as the midpoint between these two extremes. Similarly, mean slope and aspect were obtained by averaging within each polygon the DEM-derived slope and aspect values. The surface area of each rock glacier (in km²) was computed using QGIS geometry tools. The smallest rock glacier included in the inventory has an area of 0.001 km², the minimum area threshold for inclusion, according to the IPA guidelines (RGIK, 2023). The climatic variables considered were mean annual air temperature (MAAT), mean annual ground temperature (MAGT), annual precipitation (AP), and potential incoming solar radiation (PISR). These data were obtained by overlaying the rock glacier polygons on the respective climate raster layers and extracting the mean value of all pixels within each polygon for each variable. Each rock glacier was thus attributed with an average MAAT, MAGT, AP and PISR based on its location. All the above topographic and climatic parameters were compiled into the rock glacier inventory's attribute table (Table 3).

## 4.6 Uncertainty assessment

Mapping and interpreting rock glaciers present many challenges due to the inherently subjective nature of optical image analysis. This subjectivity can lead to significant variability in delineated rock glacier boundaries, as demonstrated by Brardinoni et al. (2019), who observed considerable differences when multiple analysts mapped the same rock glaciers on high-resolution imagery. To reduce mapping subjectivity and quantify the uncertainty in the rock glacier area estimates, an intercomparison experiment was carried out. Five different analysts independently mapped the outlines of a representative set of 400 rock glaciers (Fig. 4). Then the uncertainty in area for each rock glacier was quantified by calculating the standard deviation of the five area values obtained.



In addition, how uncertainties in delineation translate into uncertainties in derived properties such as elevation and slope was assessed. For each rock glacier in the intercomparison set, we compared the minimum elevation, maximum elevation, and mean slope values derived from each analyst's mapping. The discrepancies in these parameters were analyzed using Bland–Altman plots (Bland and Altman, 1986) to evaluate any systematic biases or limits of agreement with adjustments for spatial data.

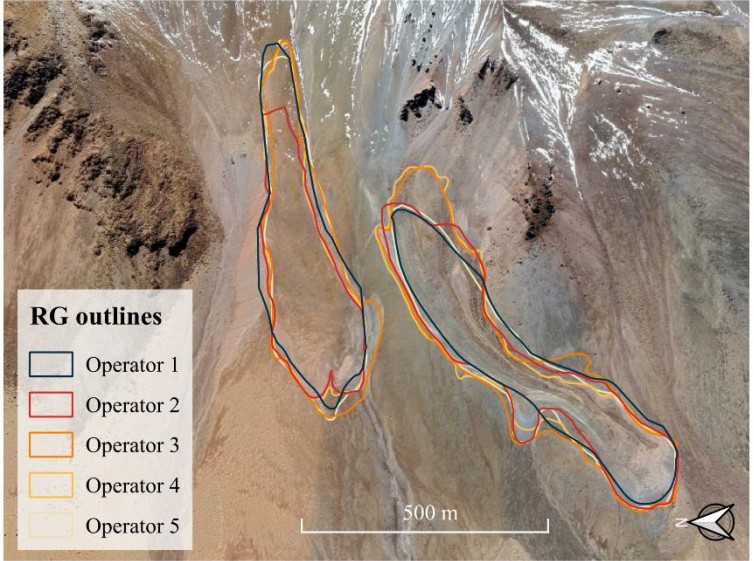


**Figure 4.** Example of mapping variability: outlines of the same rock glacier delineated by five different analysts are superimposed. Each colored outline represents one analyst's mapping of the same landform. Basemap: © Microsoft Corporation. Bing Maps imagery (2024).

## 5 Results

### 5.1 Rock glacier inventory overview

In total, 2338 rock glaciers were identified in the Peruvian Andes, with a total mapped area of $94.09 \pm 0.05$ km². The size of individual rock glaciers ranges from very small features (~0.001 km²) to large complexes (~0.83 km²), with an average area of ~0.04 km² per rock glacier. Spatially, the distribution of rock glaciers is highly uneven across the country. The vast majority are located in the Southern Dry Outer Tropics (SDOT) subregion. Specifically, 2135 rock glaciers (~91 % of the total) were mapped in SDOT, indicating the southern Peruvian Andes as the principal rock glacier zone. The Southern Wet Outer Tropics

(SWOT) contains 139 rock glaciers (~6 %), while the northern subregions – Northern Dry Outer Tropics (NDOT) and Northern Wet Outer Tropics (NWOT) – contain only 17 (~1 %) and 47 (~2 %) rock glaciers, respectively. This stark contrast underscores the influence of the drier, colder climate in southern Peru compared to the more tropical (warmer/wetter) north. Lithology is also likely a key factor. The highest percentage of inventoried rock glaciers coincides with volcanic rock outcrops, where the

type of chemical alteration increases the albedo of the surfaces and enhances permafrost development and preservation
(Yoshikawa et al., 2020).

The inventory spans 35 distinct river basins across the Andes of Peru. There are 4 basins in NDOT, 5 in NWOT, 19 in SDOT, and 7 in SWOT that contain at least one rock glacier. Even within the rock glacier-rich south, their distribution varies by basin. For example, the Camaná basin (in SDOT) has the highest number of rock glaciers (507 features) covering a total area of ~17.43 km², with a mean rock glacier elevation around 5005 m a.s.l. In contrast, the neighboring Ocoña basin (also SDOT)
contains slightly fewer rock glaciers (463), but those tend to be larger, yielding a greater total rock glacier area (~23.08 km²) at a similar mean elevation (~5001 m a.s.l.).

This suggests that even basins in close proximity, under the same broad climatic subregion, can exhibit different rock glacier densities and size distributions – possibly due to local geomorphological factors (such as basin lithology or glacial evolution). Generally, basins in the outer tropics of southern Peru (SDOT and SWOT) host dozens to hundreds of rock glaciers, whereas
basins in the central and northern Andes host only a handful at most. Table 5 provides a detailed breakdown of rock glacier counts and areas by subregion and river basin.

**Table 5.** Rock glaciers by subregion and basin.

| Subregion | Tributary River Basin | Count | Area (km²) | Mean Elevation (m a.s.l.) |
|---|---|---|---|---|
| NDOT | Cuenca Mala | 8 | 0.19 | 4879 |
| | Cuenca Rímac* | 4 | 0.11 | 4989 |
| | Cuenca Pativilca* | 3 | 0.31 | 4820 |
| | Cuenca Chancay - Huaral | 2 | 0.02 | 4904 |
| | **All** | **17** | **0.63** | **4873** |
| NWOT | Cuenca Rímac* | 19 | 0.59 | 4858 |
| | Cuenca Cañete | 10 | 0.20 | 4883 |
| | Cuenca Pativilca* | 10 | 0.93 | 4914 |
| | Cuenca Mantaro | 4 | 0.14 | 4910 |
| | Cuenca Pampas | 4 | 0.03 | 4884 |
| | **All** | **47** | **1.89** | **4835** |
| SDOT | Cuenca Camaná | 507 | 17.43 | 5005 |
| | Cuenca Ocoña* | 463 | 23.08 | 5001 |
| | Cuenca Tambo | 238 | 8.84 | 4982 |
| | Cuenca Quilca - Vitor - Chili | 182 | 8.42 | 5021 |
| | Cuenca Locumba | 130 | 6.24 | 5040 |



| | | | | |
|---|---|---|---|---|
| | Cuenca Ilo - Moquegua | 116 | 4.15 | 4976 |
| | Cuenca Mauri | 116 | 4.09 | 4965 |
| | Cuenca Sama | 115 | 5.58 | 5007 |
| | Cuenca Ushusuma | 70 | 2.91 | 5103 |
| | Intercuenca Alto Apurímac | 65 | 2.17 | 4929 |
| | Cuenca Ilave | 53 | 1.70 | 4978 |
| | Cuenca Caplina | 50 | 1.94 | 5068 |
| | Cuenca Caño | 14 | 0.80 | 5035 |
| | Cuenca Cañete | 6 | 0.12 | 4890 |
| | Cuenca Lluta | 4 | 0.15 | 5217 |
| | Cuenca San Juan | 2 | 0.03 | 4980 |
| | Cuenca Yauca | 2 | 0.04 | 4870 |
| | Cuenca Pescadores - Caraveli | 1 | 0.02 | 4595 |
| | Cuenca Pisco | 1 | 0.01 | 4642 |
| | **All** | **2135** | **87.73** | **5025** |
| SWOT | Cuenca Urubamba | 40 | 0.89 | 5006 |
| | Intercuenca Alto Apurímac | 37 | 1.42 | 4973 |
| | Cuenca Azángaro | 32 | 0.82 | 5017 |
| | Cuenca Pucará | 14 | 0.35 | 4917 |
| | Cuenca Coata | 11 | 0.18 | 4932 |
| | Cuenca Suches | 3 | 0.08 | 5004 |
| | Cuenca Ocoña* | 2 | 0.11 | 5095 |
| | **All** | **139** | **3.84** | **4937** |

Note: 'Cuenca' means 'basin' in Spanish. *Basin spans multiple subregions; counts given per subregion portion

Among the 2338 rock glaciers, 1790 are smaller than 0.1 km$^2$, 517 are in the 0.10 to 0.20 km$^2$ range, and 27 are between 0.20
to 0.40. Only 4 rock glaciers exceed 0.4 km$^2$, including one that exceeds 0.6 km$^2$; these larger glaciers represent approximately
0.17 % of the total area of rock glaciers located in the Peruvian Andes. The rock glacier size category of <0.10 km$^2$ covers the
largest area with 76.6 %, while the 0.1 to 0.2 km$^2$ category is the second largest category (22.1 %), while only 1.15 % of rock
glaciers are between 0.2 to 0.4 km$^2$.  Rock glaciers with an area <0.10 km$^2$ are found at a relatively low mean altitude near
4990 m a.s.l., with the category between 0.4 to 0.6 presenting the lowest mean altitude of 4936 m a.s.l., while the other three
categories have similar mean altitudes of 5026 m a.s.l. (0.1 to 0.2 km$^2$), 5057 m a.s.l. (0.2 to 0.4 km$^2$) and 5081 m a.s.l. (>0.6





km$^2$). In addition, rock glaciers less than 0.10 km$^2$ in area have a slightly higher mean slope of ~21° compared to a mean slope between 14.69 and 18.52° in the other four size categories (Table 6).

**Table 6.** Size categories of rock glaciers.

| Size category (km$^2$) | Area (km$^2$) | | | Mean Elevation (m a.s.l.) | Mean Slope (°) |
| --- | --- | --- | --- | --- | --- |
| | Count | Mean | Total | | |
| < 0.1 | 1790 (76.60 %) | 0.02 | 40.23 ± 0.01 | 4990 ± 169 | 21.38 |
| 0.1 – 0.2 | 517 (22.10 %) | 0.09 | 44.98 ± 0.03 | 5026 ± 194 | 18.52 |
| 0.2 – 0.4 | 27 (1.15 %) | 0.25 | 6.78 ± 0.05 | 5057 ± 277 | 15.47 |
| 0.4 – 0.6 | 3 (0.13 %) | 0.43 | 1.28 ± 0.02 | 4936 ± 214 | 14.69 |
| > 0.6 | 1 (0.04 %) | 0.83 | 0.83 | 5081 | 14.93 |

**5.2 Rock glacier classification**

Morphological types: the mapped rock glaciers exhibit various plan-view shapes. A majority (~55 %) of the inventoried rock glaciers are tongue-shaped, meaning they have a single elongate lobe extending downslope. About 26 % are lobate forms, which are wider and shorter (often at the base of escarpments). Another 11 % have polymorphic or complex shapes (e.g., coalesced multi-lobate forms), and the remaining ~8 % are classified as coalescent rock glaciers (where two or more units merge into a combined body). This morphological categorization helps to describe the inventory's diversity: tongue-shaped

rock glaciers dominate, but a significant fraction shows evidence of multiple source areas or unusual shapes due to local topography.

**5.2.1 Rock glacier activity:**

The inventory contains 715 active rock glaciers, 1162 transitional, and 461 relics. These correspond to roughly 31 % active, 49 % transitional, and 20 % relict of the total population (see Table 7 for summary). The active rock glaciers (those presumed

to still contain ice and exhibit creep) are primarily found at the highest elevations and often in the southern subregions. Transitional rock glaciers (which likely have degrading ice) form the largest category, nearly half of all cases, indicating widespread permafrost degradation or marginal activity. Relict rock glaciers (ice-free and inactive) make up about one-fifth of the inventory, typically occurring at the lower elevations of the rock glacier range or in slightly warmer locales.

**Table 7.** Summary of characteristics by activity status.

| Activity | Area (km$^2$) | | | | | Elevation (m a.s.l.) | | | Mean Slope (°) |
| --- | --- | --- | --- | --- | --- | --- | --- | --- | --- |
| | Count | Min | Max | Mean | Total | MAF | Max Elev | Mean Elev | |
| Active | 715 (30.58 %) | 0.003 | 0.42 | 0.06 | 39.75 ± 38.01 | 5001 ± 212 | 5134 ± 207 | 5065 ± 203 | 21.19 |
| Transitional | 1162 | 0.002 | 0.83 | 0.04 | 40.93 ± 47.60 | 4945 ± 151 | 5032 ± 155 | 4986 ± 152 | 20.49 |





|  |  |  |  |  |  |  |  |  |  |
|---|---|---|---|---|---|---|---|---|---|
|  | (49.70 %) |  |  |  |  |  |  |  |  |
| Relict | 461 (19.72 %) | 0.001 | 0.44 | 0.03 | 13.41 ± 17.61 | 4894 ± 159 | 4970 ± 159 | 4930 ± 157 | 20.29 |

In terms of area, the active rock glaciers account for ~42.0 % of the total mapped rock glacier area (39.75 km² out of 94.09 km²). Transitional rock glaciers make up about 43.5 % of the area (approximately 40.9 km²), and relict rock glaciers constitute the remaining ~14.5 % (~13.4 km²). These figures show that active and transitional rock glaciers, while roughly equal in count, dominate the area because relict rock glaciers tend to be smaller on average. Indeed, we found that relict rock glaciers have the smallest mean size (mean area ~0.03 km²), compared to ~0.04 km² for transitional and ~0.06 km² for active rock glaciers. This pattern is consistent with expectations: once a rock glacier loses its ice (becoming relict), it may slump and shrink over time, whereas active ones are buttressed by internal ice and can maintain larger extents.

Spatial differences in activity/morphology: the prevalence of active vs. relict rock glaciers varies by subregion. Figure 5 shows that the SDOT subregion contains by far the most rock glaciers in all categories (active, transitional, relict), with transitional forms slightly outnumber active and relict being least common. This aligns with SDOT's arid climate that supports many rock glaciers, some of which are in advanced stages of development. The SWOT subregion (more humid) has fewer total rock glaciers, and transitional types dominate there – likely reflecting that in a wetter climate, only the most favorably located rock glaciers remain active, while others are degrading. he NWOT and NDOT subregions in northern Peru have very few rock glaciers overall; NDOT in particular (hyper-arid but warmer) shows the lowest counts, and those few tend to cluster near the highest peaks.

Interestingly, the Minimum Altitude of rock glacier Fronts (MAF) is relatively uniform across subregions – differing by only on the order of 100 m. In both northern and southern subregions, active rock glacier fronts are found around 4800–5000 m a.s.l. This suggests that while climate aridity is a big driver of rock glacier abundance, the elevation of the permafrost limit (and hence where active rock glaciers can exist) is primarily controlled by temperature (which correlates with latitude and elevation). In other words, even though SDOT has many more rock glaciers than NDOT/NWOT, the lowest elevation at which active ones occur is only slightly higher in SDOT, implying a latitude effect (southern Peru is a bit cooler, allowing rock glaciers to exist a tad lower there) superimposed on the overall tropical environment.

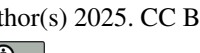



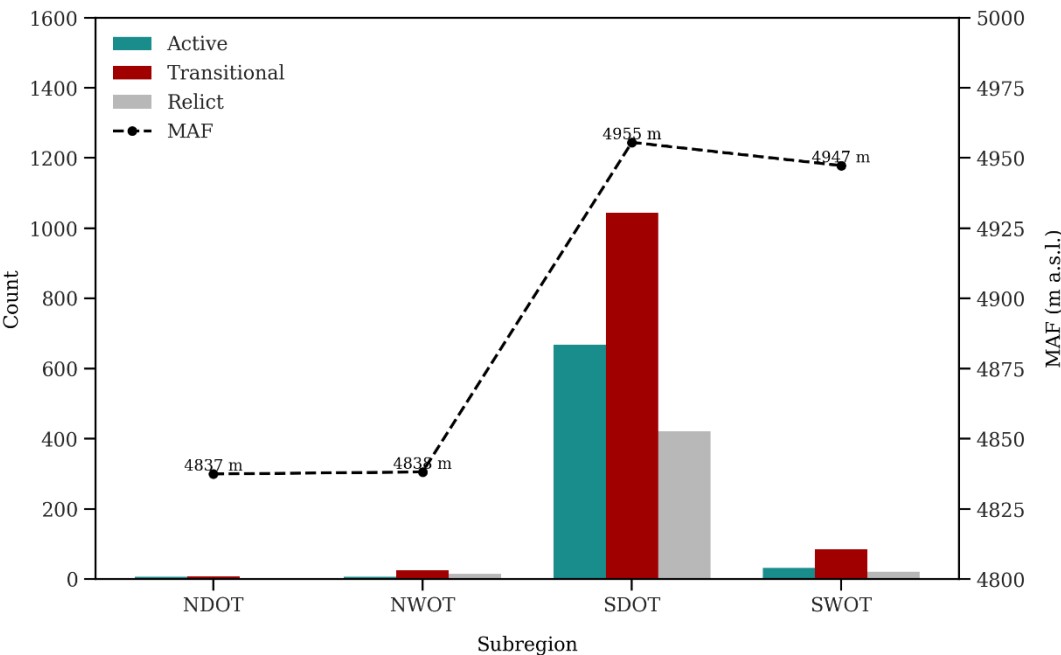

**Figure 5.** Rock glacier abundance by activity status and subregion. MAF values indicate the lowest elevation of rock glacier fronts by subregion.


### 5.3. Topoclimatic characterization

Elevation distribution: the rock glaciers in the inventory are roughly between 4400 m and 5800 m a.s.l. The average minimum elevation (i.e., the mean of each rock glacier's lowest point) is ~4951 m, while the average maximum elevation (mean of highest point of each rock glacier) is ~5051 m. The vast majority of rock glaciers are situated in a relatively narrow elevational

band: about 75 % of the inventory lies between 4800 m and 5200 m a.s.l (Fig. 6). In contrast, only ~10.5 % of rock glaciers have elevations dropping into the 4400–4800 m range (usually these would be relict or very marginal features), and ~14.2 % reach elevations of 5200–5800 m (often active ones in the highest peaks). The total area covered by rock glaciers is very similar in the 4800–5000 m band (about 36.0 km²) and the 5000–5200 m band (about 34.8 km²). This indicates that while there are a greater number of rock glaciers in the 4800-5000 m band, the somewhat less numerous rock glaciers above 5000 m tend to be

larger in size, which balances the total area covered by rock glaciers in each of the bands. Very few rock glaciers are found below ~4400 m, consistent with the current permafrost limit (Yoshikawa et al., 2020), and none of the active ones go below ~5000 m as noted earlier.

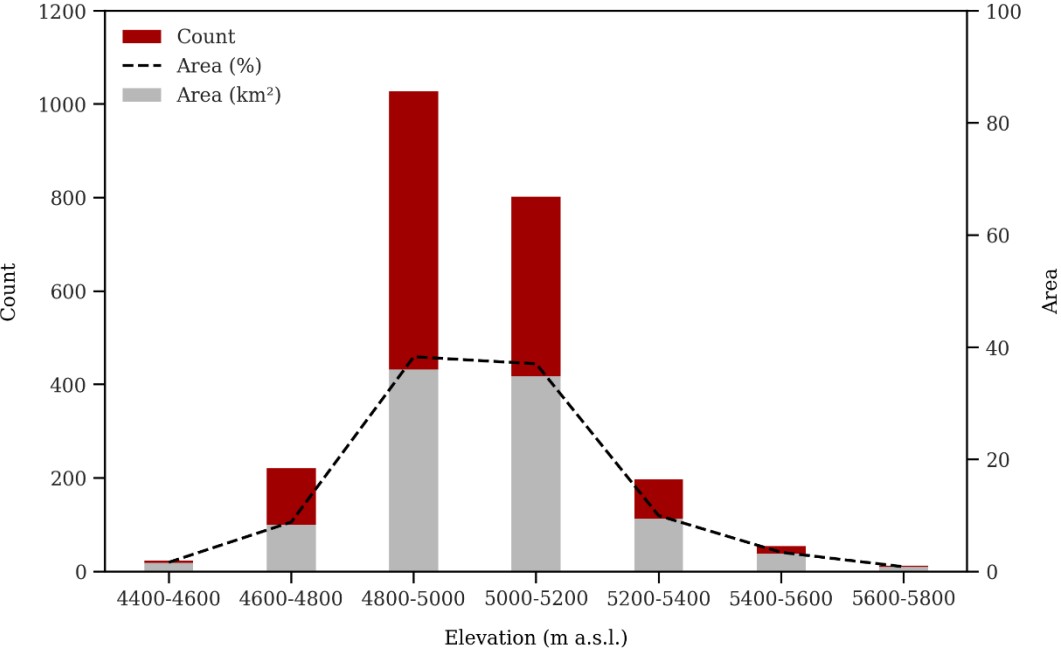

Figure 6. Elevation distribution of rock glaciers. The histogram shows the number of inventoried rock glaciers within 200-meter elevation intervals (based on mean elevation of each feature).

### 5.3.1 Slope distribution

The mean slope of rock glacier surfaces ranges from gentle (~7°) to quite steep (~37–38°) across the inventory. The average mean slopes is ~21°, indicating that most rock glaciers lie on moderate slopes. We found that the 15–20° slope class contains
the largest number of rock glaciers (891 features) and the largest share of total rock glacier area (~44 %). This suggests that slopes around 15–20° are especially conducive to rock glacier formation/preservation, likely because they are steep enough for debris-ice creep but not so steep as to cause frequent avalanching or debris removal. The next most populated slope class is 20–25° (779 rock glaciers, contributing ~29 % of total area) and then 25–30° (354 rock glaciers, ~8 % of area). Very steep mean slopes (>30°) are less common – rock glaciers tend to not maintain such steep overall slopes, though their fronts might
be that steep. These findings (illustrated in Fig. 7) imply that Andean rock glaciers typically form on moderate slopes, which is consistent with global observations of rock glacier topography.

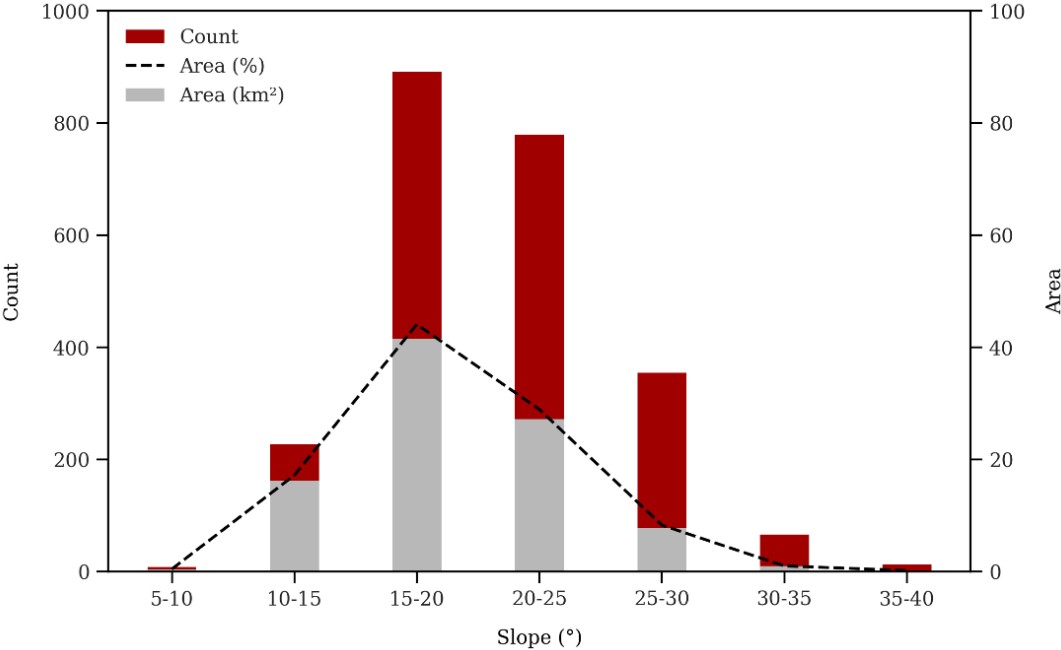

**Figure 7.** Rock glacier frequency and area by slope. The inventory is binned into slope angle categories (e.g., 5–10°, 10–25°, etc.), and the plot shows both the number of rock glaciers (left axis) and the total area (right axis) in each slope bin.

### 5.3.2 Aspect (orientation)

Aspect analysis confirms a strong preferential orientation of rock glaciers toward the southern half of the compass. Approximately 89.5 % of all inventoried rock glaciers have aspects between east-southeast and west-southwest. The most frequent orientations are due to south (~36 %), southeast (~28 %), and southwest (~19 % of rock glaciers) as shown in Fig. 8. This bias toward southerly orientations is expected in the Southern Hemisphere tropics, as south-facing slopes receive less solar radiation and thus maintain colder ground conditions favourable for permafrost. Only ~5.0 % of rock glaciers face predominantly northward (including N, NE, NW categories combined). Pure north-facing rock glaciers are extremely scarce (<1 % of the population), and even northeast (2 %) or northwest (1 %) facing ones are rare. A few exceptions exist in specific local microclimates – for instance, in some deep valleys or shaded cirques in NWOT or SWOT, a limited number of rock glaciers have NE aspects. However, even those "anomalously" oriented rock glaciers still tend to have some mitigating factors (such as shading from nearby peaks) that allow them to persist despite more sun exposure. Overall, the aspect analysis underscores that solar radiation (and by proxy, aspect) is a crucial control on rock glacier presence in the tropics, second only to elevation/temperature.

Rock glaciers in the Peruvian Andes show different climatic dependencies between subregions, predominantly clustering within a MAAT range of -2 to 4 °C despite encompassing precipitation gradients of 500 to 2500 mm/yr. The SDOT subregion shows the highest density of rock glaciers with intermediate precipitation (500 to 1000 mm/yr), while other subregions show





divergent patterns: NDOT and NWOT show predominantly occurrences of MAAT >0 °C (including active forms), and SWOT reaches maximum precipitation (1500 to 2500 mm/yr) without thermal extremes. Notably, transitional forms of SDOT dominate the entire MAAT spectrum, whereas relict glaciers are concentrated above 0 °C (Fig. 9). This distribution reveals

that elevation is the primary control on the presence of rock glaciers, with secondary hydrothermal modulation - evidenced by inverse AP-MAAT relationships in NDOT/SDOT/SWOT (where aridity and snow redistribution at elevation increase cooling) versus the direct correlation of NWOT (driven by thermal depression mediated by microclimatic processes). Such systematic variations underscore how Andean rock glaciers integrate macroscale climatic gradients with local topoclimatic processes.

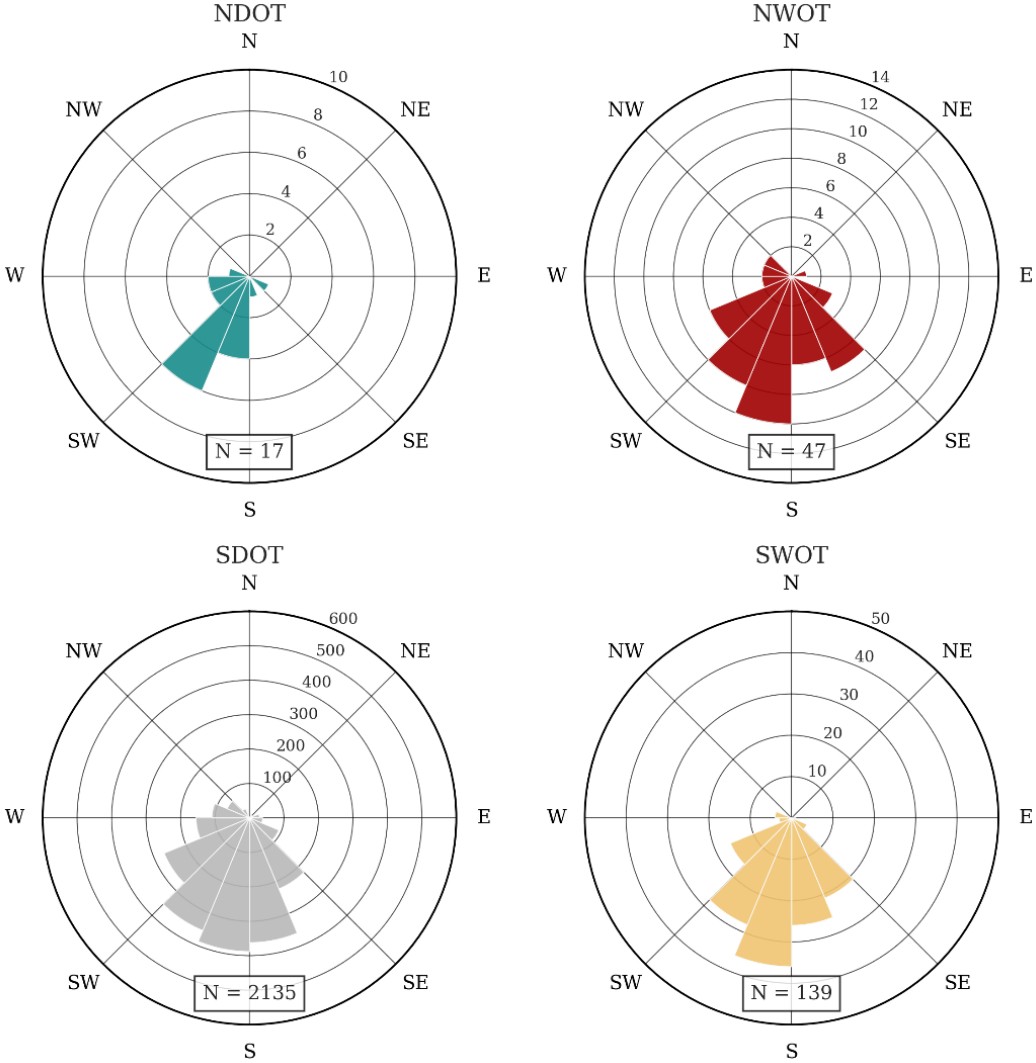

**Figure 8.** Aspect distribution of rock glaciers by subregion. For each subregion (NWOT, NDOT, SWOT, SDOT), a rose diagram of rock glacier aspect is shown, indicating a strong preference for south-facing orientations.



**Figure 9.** Spatial distribution of rock glaciers analyzed according to Annual Precipitation (AP), Mean Annual Air Temperature (MAAT), Potential Incoming Solar Radiation (PISR), and Area in the four subregions.



## 5.4 Uncertainty analysis

The mapping uncertainty of the inventory was evaluated by comparing results among different mappers and considering potential delineation errors. The area of each rock glacier polygon comes with an uncertainty estimate based on multi-analyst mapping trials. Individual rock glacier area uncertainty was found to range from as low as ~0.001 km² for small, clearly defined features to up to ~0.3 km² for very large or diffuse features, with a mean uncertainty of ~0.06 km². This translates to an overall relative uncertainty of about 18 % in area on average. Notably, this level of uncertainty is lower than values reported in some other manual rock glacier inventories (e.g., those compiled over larger regions or by less experienced mappers, see Brardinoni et al., 2019), indicating the robustness of this methods.

To further quantify consistency, a Bland–Altman analysis was performed (see Fig. S1 in the Supplement) on a subset of rock glaciers independently mapped by five different team members. The results showed very high agreement: the mean difference (bias) in mapped area between operators was close to zero (dashed red line in Fig. S1), and the 95 % limits of agreement (±1.96 SD, dashed gray lines) were within acceptable ranges, even for the largest rock glaciers. For example, the average discrepancy in calculated mean elevation for a rock glacier between mappers was on the order of 10 m (0.2 %), and for mean slope it was ~1.5° (4 % variance). The Intraclass Correlation Coefficients (ICC) for key attributes (area, elevation, slope) all exceeded 0.95, which confirms excellent consistency among mappers. A slight heteroscedastic trend was observed: larger rock glaciers showed proportionally a bit more variance in area mapped (as noted by a conical distribution in Fig. S1), but the effect is minor.

Overall, this uncertainty analysis gives confidence that errors in mapping are small relative to the size of landforms, and that our inventory is reproducible. The careful quality control (Section 4.1.3) further reduced uncertainty by removing or correcting dubious polygons. Some degree of subjectivity is inevitable in manual digitizing, but by documenting these uncertainties, transparency is achieved. Future work could apply semi-automated mapping or higher-resolution data to further reduce uncertainty, but for a national-scale manual inventory, an ~18 % area uncertainty is within acceptable bounds and substantially improves upon previous knowledge, where uncertainties were effectively 100 % in unmapped regions.

The results provide the first comprehensive picture of rock glaciers in Peru. Below their significance in the context of regional permafrost and prior studies is discussed.

## 6 Discussion

### 6.1 Rock glacier distribution in Peru

The primary objective of this work was to create a comprehensive, geospatially explicit inventory of rock glaciers in the Peruvian Andes, and to analyze their characteristics and controlling factors. The resulting dataset provides a much-needed baseline for understanding mountain permafrost in tropical South America, with implications for hydrology, climate impact studies, and natural hazard assessments. A total of 2338 rock glaciers have been catalogued, of which 715 (30.6 %) are active,



1162 (49.7 %) transitional, and 461 (19.7 %) relict – figures that underscore a substantial permafrost presence in Peru's high mountains. The strong clustering of rock glaciers in the southern (SDOT) subregion of Peru points to the importance of regional climate: the southern Peruvian Andes are more arid and slightly cooler, providing favorable conditions for rock glacier formation, whereas the northern and central Andes (wetter and/or warmer) host very few permafrost landforms. This mirrors

observations from adjacent countries – for instance, in Bolivia and northern Chile, rock glaciers are far more abundant in the dry Andes than in the wet, inner-tropical Andes (Rangecroft et al., 2014; Perucca and Esper Angillieri, 2011). However, the role of high albedo stemming from volcanic rock surface weathering cannot be dismissed as a key factor (Yoshikawa et al., 2020). The inventory confirms that Peru fits this broader pattern, with a pronounced gradient in rock glacier frequency from south to north.

The average sizes of Peruvian rock glaciers appear somewhat smaller than those reported in mid-latitude Andes inventories. For example, we found mean areas on the order of 0.06 km² for active and 0.03 km² for relict rock glaciers, whereas Rangecroft et al., (2014) reported larger averages (~0.13 km² active, 0.10 km² relict) for the Bolivian Andes. Similarly, rock glaciers in the Patagonian Andes tend to be larger (mean ~0.09 km² for intact ones; Falaschi et al., 2015). An inventory in Argentina's semi-arid Andes (Falaschi et al., 2016) found mean areas of ~0.07, 0.06, and 0.12 km² for active, transitional, and relict

categories, respectively. In our dataset, relict forms are the smallest on average, whereas in that Argentine inventory relict forms were relatively large – possibly indicating different evolutionary stages or detection biases. These discrepancies may arise from a combination of factors: (i) climatic differences (tropical alpine permafrost may produce smaller rock glaciers due to higher snowfall and thus, more glacier-dominated past conditions , whereas drier climates yield larger rock glaciers); (ii) geological/lithological/climatic differences affecting debris supply; and (iii) methodological differences (e.g., the resolution

of imagery and the criteria for defining polygon extents). It is worth noting that our use of very high-resolution imagery and careful delineation might result in more conservative area estimates compared to coarser inventories. Overall, the smaller size of Peruvian rock glaciers is not unexpected given the extreme elevation and tropical conditions: only the most resilient landforms persist, and many may represent the remnants of once larger features.

A critical metric for comparing permafrost environments is the mean altitude of the rock glacier fronts (MAF), which serves

as a proxy for the lower limit of discontinuous permafrost in a region. In our inventory, the mean front altitude for active rock glaciers is ~5001 m a.s.l., which aligns closely with independent estimates of the 0 °C MAAT isotherm in the Peruvian Andes (around 5000–5030 m according to León et al., 2021). This concurrence confirms that active rock glacier fronts indeed mark the current permafrost boundary.

We observed no active rock glaciers below ~5000 m, reinforcing that ~5000 m a.s.l. is essentially the climatic permafrost limit

in Peru at present. Interestingly, this limit appears slightly lower (by on the order of 50–100 m) than what has been reported in the Cordillera Volcánica of southern Peru (Badillo-Rivera et al., 2021) found active rock glaciers down to ~5080 m, which we slightly exceed) and the Bolivian Andes (~4980 m; Rangecroft et al., 2014), but it is dramatically higher than the permafrost





limits in the extra-tropical Andes – for instance, ~2660 m in the semi-arid Argentine Andes, and ~1760 m in Patagonia (Falaschi et al., 2016; Selley et al., 2028).

These contrasts illustrate the dominant effect of latitude on permafrost elevation: the tropics require much greater elevations to sustain permafrost due to higher air temperatures and intense solar radiation. The fact that Peru's rock glaciers are among the world's highest-elevation rock glaciers is consistent with this reasoning.

The comparisons with global inventories (see Table 8) show that while Peru's rock glaciers are extreme in elevation, other characteristics like slope and aspect are broadly similar to rock glaciers elsewhere. The mean slope (~20–21°) we found is in

line with typical values reported in the Alps or Himalaya (Robson et al., 2020). Likewise, the preference for pole-facing aspects is a universal trait (Northern Hemisphere rock glaciers preferentially face north, Southern Hemisphere face south). Thus, Peruvian rock glaciers conform to global patterns in those regards, strengthening confidence that we are observing the same periglacial processes albeit in a tropical setting.

**Table 8.** Characteristics of rock glacier inventories reported from various regions of the world.

| Study area | Count | Area (km²) | Activity* | Min. Elevation (m a.s.l.) | Max. Elevation (m a.s.l.) | Mean Elevation (m a.s.l.) | Source |
|---|---|---|---|---|---|---|---|
| **South America** | | | | | | | |
| Central Patagonian | 89 | 14.18 | Intact | 1766 | 1941 | - | Selley et al. (2018) |
| | | | Relict | 1758 | 1919 | - | |
| Aconcagua River Basin | 669 | 70.00 | All | 2370 | 4565 | 3810 | Janke et al. (2017) |
| Central Andes, Atacama region | 477 | 44.34 | All | 3807 | 5504 | 4427 | García et al. (2017) |
| Volcán Domuyo region, Southern Central Andes | 224 | 17.70 | Active | 2664 | 3968 | 3047 | Falaschi et al. (2016) |
| | | | Transitional | 2165 | 3526 | 2821 | |
| | | | Relict | 1955 | 3340 | 2644 | |
| Monte San Lorenzo massif | 177 | 11.31 | Intact | 1335 | 2155 | 1742 | Falaschi et al. (2015) |
| | | | Relict | 1267 | 2030 | 159 | |
| Valles Calchaquíes region | 488 | 58.50 | Intact | 4183 | 5908 | 4873 | Falaschi et al. (2014) |
| | | | Relict | 4072 | 5397 | 4695 | |
| Huasco Valley, Chile | 50 | 8.60 | All | 3840 | 5070 | 4220 | Hess et al. (2020) |
| Bolivian Andes | 94 | 0.24 | Active | 4983 | 5162 | - | Rangecroft et al. (2014) |
| | | | Relict | 4870 | 5009 | - | |
| Argentinean Andes | 8454 | 952 | Active | 3951 | 4102 | 4018 | IANIGLA (2018) |
| | | | Transitional | 3865 | 3986 | 3919 | |
| | | | Intact | 3935 | 4718 | 4274 | |
| Chilean Andes | 3586 | 483 | All | 3801 | 4037 | 3903 | DGA (2022) |
| Cordillera Volcánica, Perú | 187 | 8.30 | Active | - | - | 5081 | Badillo-Rivera et al. (2021) León et al. (2021) |
| | | | Transitional | - | - | 4961 | |
| | | | Relict | - | - | 4851 | |
| Peruvian Andes | 2338 | 94.09 | Active | 5001 | 5134 | 5065 | This study |
| | | | Transitional | 4945 | 5032 | 4986 | |
| | | | Relict | 4894 | 4970 | 4930 | |
| **North America** | | | | | | | |





| Colorado Front Range | 220 | 19.90 | Active | 3525 | 3668 | 3594 | Janke (2007) |
|---|---|---|---|---|---|---|---|
|  |  |  | Transitional | 3424 | 3541 | 3477 |  |
|  |  |  | Relict | 3227 | 3358 | 3288 |  |
| **Europe** | | | | | | | |
| Western Carpathian | 383 | 13.84 | Intact | 1911 | - | - | Uxa and Mida (2017) |
|  |  |  | Relict | 1688 | - | - |  |
| Southern Carpathian | 306 | 12.70 | All | - | - | 1998 | Onaca et al. (2017) |
| Southern region of eastern Italian Alps | 705 | 33.30 | Intact | 2716 | 3082 | 2632 | Seppi et al. (2012) |
|  |  |  | Relict | 1644 | 2669 | 2169 |  |
| South-eastern Alps | 53 | 3.45 | All | - | - | 1778 | Colucci et al. (2016) |
| Tyrolean Alps | 3145 | 167.2 | Active | 2628 | 2797 | 2704 | Krainer and Ribis (2012) |
|  |  |  | Transitional | 2542 | 2665 | 2598 |  |
|  |  |  | Relict | 2279 | 2384 | 2330 |  |
| **Asia** | | | | | | | |
| Tibetan Plateau | 44273 | 6000 | All | 4000 | 5000 | 4729 | Sun et al. (2024) |
| Kashmir Himalaya | 240 | 41.25 | Active | 3481 | 4485 | 4015 | Abdullah and Romshoo (2024) |
|  |  |  | Transitional | 3672 | 4349 | 4010 |  |
|  |  |  | Relict | 3316 | 3939 | 3650 |  |
| Western Himalaya, India | 5807 | 712 | All | 3190 | 5753 | 4491 | Bhat et al. (2025) |
| Southwestern Pamirs | 275 | 55.52 | Active | 4291 | 4593 | 4442 | Ma and Oguchi (2024) |
|  |  |  | Transitional | 4342 | 4541 | 4441 |  |

*"Inactive" class is the same to "Transitional" class and "Intact" class include "Active" and "Transitional (Inactive)" classes.

Nevertheless, this dataset is subject to several limitations. Factors such as seasonal snow cover, persistent cloudiness, and topographic shadowing, especially near ice glaciers or in high-relief areas, hindered consistent visual interpretation and may have led to the omission of some features. Although manual verification and cross-checks were conducted, the possibility of

omissions due to image limitations or mistakes in the human interpretation cannot be fully excluded (Brardinoni et al., 2019; Pandey, 2019). Despite these limitations, the dataset presented here represents a significant advancement in the mapping and understanding of tropical periglacial landscapes. It offers a consistent and reproducible foundation for the study of cryospheric processes in the Peruvian Andes and supports transdisciplinary applications in water resources, ecosystem services, natural hazards, and climate monitoring. The inventory's harmonized format also facilitates future regional and global-scale

comparisons, as well as integration into modeling frameworks assessing the role of rock glaciers in mountain hydrology under changing climatic conditions.

**6.2 Environmental controls on rock glaciers**

The analysis of this inventory highlights climate (temperature and precipitation) as the overarching control on rock glacier distribution in Peru. Rock glaciers are essentially absent in areas with higher mean annual air temperatures (MAAT) and

abundant precipitation, such as the wetter northern Andes, and are prolific in cold, arid areas, such as the southern outer tropics.



Low MAAT ensures the presence of permafrost, and low precipitation (with associated thin snow cover) reduces insulation of the ground and allows deeper winter cooling – both conditions are necessary for rock glaciers to develop and persist (Esper Angillieri, 2017).

In the subregional breakdown, the SDOT has the lowest average MAAT and a pronounced dry season, aligning with its 545 dominance in rock glacier count. By contrast, NWOT and parts of SWOT have higher precipitation and more cloud cover, which likely limit permafrost despite some high elevations; accordingly, rock glaciers there are few or marginal (often transitional or relict).

An interesting observation is that within the zones where rock glaciers do exist, microclimatic variations (like slope aspect and shading) further control where active versus relict forms are found. For instance, in SWOT (southern, wet), rock glaciers tend 550 to survive only on south-facing, well-shaded slopes at the highest elevations – conditions that locally offset the warmer/wetter climate. Transitional and relict rock glaciers in SWOT and NWOT indicate that permafrost likely existed more broadly during past colder periods (e.g., the Little Ice Age), but current conditions are only just sufficient to maintain a few active ones. In SDOT, even north-facing slopes largely lack rock glaciers, despite dryness, because solar radiation is too high – again underscoring that solar aspect is a key factor.

The correlation of the inventory with modelled MAGT data (Obu et al. 2019) provides an independent check: nearly all active rock glaciers lie in grid cells where MAGT is at or below 0°C, whereas relict rock glaciers occupy cells where MAGT is just above 0°C (indicating marginal permafrost conditions). This pattern suggests that our activity classifications are climatologically consistent. It also means that rock glaciers can be used as empirical evidence of permafrost presence in data-sparse regions (in line with Brenning, 2005): wherever active rock glaciers were mapped, permafrost likely exists, and the 560 lower limit of those occurrences closely tracks the altitude of the 0°C isotherm. Conversely, areas in Peru that are cold enough (by MAAT) but lack rock glaciers likely either do not have the right topography or lacked sufficient debris supply/glacier history to form them.

In summary, temperature (elevation/latitude) sets the broad-brush possibility for rock glaciers (tropical permafrost is restricted to >~5000 m), while precipitation and related factors fine-tune their distribution. Dry areas produce more rock glaciers, 565 probably due to lower precipitation and temperature conditions, in addition to greater freeze-thaw weathering that produces debris. Too much precipitation can mean thicker snow that insulates ground or encourages glaciers instead of rock glaciers. These findings are consistent with regional studies in South America (e.g., Azócar and Brenning, 2010; García et al. 2017) that emphasize a "sweet spot" of cold and dry conditions for rock glacier prevalence. Future climate changes – warming and shifts in precipitation – will likely have complex effects: warming alone will shrink the climatic envelope for rock glaciers 570 upward, but increased aridity (if it occurs) could offset some insulating snow effects. Our baseline data now allow such projections to be attempted quantitatively in Peru. This analysis indicates that areas characterized by lower mean annual air temperature (MAAT) and reduced precipitation are more favorable for rock glacier development, as exemplified by the SDOT subregion compared to the other three. These findings support previous research suggesting that rock glaciers are



predominantly concentrated in colder and drier environments, typically at higher elevations and latitudes (Veettil et al., 2018).

In such conditions, permafrost formation and persistence are enhanced due to reduced melting and energy input. However, the overall smaller extent of rock glaciers in this inventory relative to other Andean regions may indicate the presence of discontinuous permafrost, which tends to be constrained to favorable microclimatic conditions—especially south-facing slopes (Andrés et al., 2011). A clear latitudinal gradient is observed, with the number of rock glaciers increasing from north to south, consistent with decreasing MAAT and precipitation levels—except in the SWOT subregion. This trend aligns with the

theoretical expectation that colder and more arid climates reduce internal ice melt and preserve debris cohesion, supporting the sustained creep of rock glaciers (Jorgenson et al., 2010). In the SWOT subregion, although precipitation increases due to moisture-laden winds from the Atlantic, low temperatures still permit the existence of rock glaciers. Conversely, in the northern subregions, the dominance of clean and debris-covered glaciers (INAIGEM, 2023), along with more temperate climates and higher precipitation levels (Bonshoms et al., 2020; Sagredo and Lowell, 2012), likely inhibits widespread rock glacier

formation.

   Compared to other regions of South America, the results of this work show that Peruvian rock glaciers are found in areas with relatively higher MAAT. For instance, Azócar and Brenning (2010) report rock glaciers in the semi-arid Andes of Chile occurring under MAATs of 0.5 to 1°C, and Esper Angillieri (2017) notes −2°C in Argentina, whereas the dataset of this inventory has an average MAAT of 1.1°C. This relatively high value in Peru may be associated with the predominance of

transitional rock glaciers, which are indicative of ongoing degradation. When analyzed by activity status, active rock glaciers occur under the coldest conditions, with a mean MAAT of 1.08°C, followed by transitional (1.21°C) and relict (1.58°C) forms. This supports the notion that permafrost and internal ice can persist under slightly positive MAATs due to thermal insulation provided by the debris mantle or active layer (Barsch, 1996; Gruber and Haeberli, 2007). Precipitation values in the Peruvian Andes exceed those reported in comparable inventories. For instance, Rangecroft et al. (2014) reported 250–300 mm/year in

Bolivia, and Esper Angillieri (2017) found values as low as 6–37 mm/year in Argentina. Despite these differences, the presence of rock glaciers at much higher precipitation levels (1000–2000 mm/year) has been documented in other high mountain environments, such as the Himalayas and the Caucasus (Abdullah and Romshoo, 2024; Tielidze et al., 2023). This supports the view that MAAT is a more critical control on rock glacier distribution than precipitation (Bhat et al., 2025).

   The mean slope of rock glaciers in this study is 20.7°, with values ranging between 7° and 37°, which is consistent with those

found in the Chilean Andes (mean 20.3°, range 0–68°; Janke et al., 2017). In terms of aspect, rock glaciers predominantly face south, southwest, and southeast—orientations generally opposite the sun's path—suggesting that reduced incoming solar radiation favors permafrost stability (Janke et al., 2017; Perucca and Esper Angillieri, 2011). Due to the lower latitude, potential incoming solar radiation (PISR) values for Peruvian rock glaciers are higher than those reported for the same landforms in the French Alps and Chilean Andes (Azocar, 2013; Marcer et al., 2017), yet they are comparable to those

documented on the Tibetan Plateau (Sun et al., 2024), possibly reflecting the high elevations and steep slopes of the Peruvian Andes.





### 6.3 Inventory uncertainties and limitations

#### 6.3.1 Delineation uncertainty and detection limits

Manual mapping, while providing high detail, inherently involves some subjectivity. As demonstrated by Brardinoni et al.
(2019), different analysts can delineate boundaries with significant variability. To quantify this, our intercomparison
experiment on a representative set of 400 rock glaciers revealed an average area uncertainty of approximately 18 % per mapped
polygon. This uncertainty typically ranges from ~0.001 km² for small, well-defined features to ~0.3 km² for very large or
diffuse ones. While this level of uncertainty is relatively low for a manual mapping product and demonstrates the robustness
of our methods compared to some other inventories (e.g., Abdullah and Romshoo, 2024), it implies that very small rock glaciers
(≤0.01 km²) are near our mapping's detection limit. Consequently, it is possible that some extremely small landforms or those
heavily obscured by debris or seasonal snow were not captured.

Our rigorous quality control process (Section 4.1.3), which involved removing ~281 low-certainty polygons and adding 32
missed ones, highlights that initial mapping can lead to both over- and under-estimation if not rigorously reviewed. Despite
this, some subjective judgment calls remained, particularly in distinguishing relict rock glaciers from morainic or talus
landforms when surface patterns were faint. We primarily relied on criteria such as the presence of a pronounced front and
lateral margins but acknowledge that a "gray zone" exists in some cases. Users of the inventory should be aware that certain
transitional vs. relict classifications might require revision with future ground truth or kinematic data.

Concordance analyses using Bland-Altman plots (Figure S1) revealed a characteristic conical pattern in pairwise comparisons
of operators, indicating that inter-rater variability was proportional to the magnitude of the measurements. The differences
between operators were more pronounced in large glaciers (>0.3 km²), mainly due to divergent interpretations of composite
systems and secondary lobes. To homogenize the inventory, composite glaciers with clear boundaries (front, edges, and
distinguishable flow patterns) were identified. Although these discrepancies reflect challenges inherent in manual mapping
(Brardinoni et al., 2019), statistical analyses show general agreement: Bias <0.02 km², 95% of differences within ±0.15 km²..
This internal consistency is further supported by Intraclass Correlation Coefficients (ICC) exceeding 0.95 for key attributes
(area, elevation, slope), confirming excellent consistency among mappers. Uncertainty associated with minimum, maximum,
and mean elevation values was found to be very low, well within the limits recommended by the IPA (≤10 %). This suggests
a robust and internally consistent mapping procedure, particularly for topographic attribute extraction.

#### 6.3.2 Temporal limitations and activity classification

The PRoGI v1.0 inventory compiles rock glaciers observed in imagery primarily from 2017 through 2025 (most Bing images
~2024, Google ~2017; see Table 1 for details). Given ongoing climate warming during this period, it is possible that some
rock glaciers classified as active based purely on their surface morphology may in fact be very sluggish or already transitional
but not yet exhibiting those characteristics. Direct kinematic data (e.g., InSAR or GPS) would be necessary to definitively



confirm current activity, and such analyses are planned for future work (see Conclusions). Our classification strategy adopted a conservative approach: if there was doubt, rock glaciers were classified as transitional (or relict) rather than active. Consequently, the count of "active" rock glaciers in this inventory should perhaps be considered a minimum estimate of truly ice-rich, moving landforms in Peru at the time of imagery acquisition.

### 6.3.3 Limitations of auxiliary data

The climate variables (MAAT, AP) used from CHELSA are model-based and at 0.1° resolution, which can smooth out important microclimates. For instance, a rock glacier located in a deeply shaded valley might experience colder conditions than the grid-averaged MAAT suggests. This discrepancy could introduce some noise in the climate correlation analysis, although the broad trends observed remain robust. Similarly, the Obu et al. (2019) MAGT data at 1 km resolution may not accurately capture small cold-air pools or the precise thermal state under thick debris. While general correlations between rock glaciers and these topoclimatic variables are reported, users should exercise caution against over-interpreting the absence or presence of any single rock glacier solely based on these grid values.

### 6.3.4 Two-dimensional mapping and volumetric estimates

This inventory is inherently two-dimensional, focusing on planform outlines. Direct measurements of ice thickness or volumetric content were beyond the scope of this study. As a result, PRoGI v1.0 alone cannot provide total ice volume or water equivalent estimates. For such applications, the inventory would need to be combined with assumptions or models of ice content (e.g., utilizing the empirical 15–70 % ice by volume range; Halla et al., 2021). This is recognized as an anticipated next step and a crucial future application of the inventory, but it falls outside the current scope of this data paper.

Despite the inherent challenges and documented uncertainties, the manually curated PRoGI v1.0 stands as a reliable and robust foundation for rock glacier research in the Peruvian Andes. The detailed uncertainty assessment, including the inter-analyst variability analysis, confirms the high consistency and reproducibility of our mapping procedure, particularly for topographic attribute extraction. While some degree of subjectivity is inevitable in manual digitizing, openly documenting these uncertainties ensures transparency and guides users in appropriate data application.

In comparison to similar inventories globally (e.g., in the Alps or High Mountain Asia), this Peruvian inventory is notable as one of the few covering an entire country in the tropics. It is anticipated that this dataset will be refined as new data and methodologies emerge. For instance, future work could explore the application of semi-automated mapping or deep learning techniques, such as those employed by Sun et al. (2024) on the Tibetan Plateau, to potentially further reduce subjectivity and detect even smaller landforms. For now, PRoGI v1.0 significantly improves upon previous knowledge, providing a critical baseline with quantified uncertainty bounds upon which future improvements can be built.





## 6.4 Significance of the inventory and future work

To our knowledge, PRoGI v1.0 is the most extensive rock glacier inventory compiled in the tropical Andes to date, and it is comparable in scope and detail to state-of-the-art inventories in other regions. By adhering to uniform mapping standards, it
provides a coherent dataset that researchers and policymakers can use with confidence. The inventory not only fills a data gap for Peru but also serves as a benchmark for future monitoring: for instance, it establishes baseline extents and classifications against which changes (e.g., rock glacier retreat or activation/inactivation) can be measured in coming decades. One important implication of this work is the integration of Peruvian rock glaciers into global networks. Because we followed the IPA Rock Glacier Inventory (RGI) scheme, our data can feed into the Global Terrestrial Network for Permafrost (GTN-P) and related
efforts to map permafrost features worldwide. his opens opportunities for interhemispheric-scale analyses, such as comparing rock glacier density and characteristics between the Andes and the Alps or Himalaya, to see how different climates yield different permafrost responses. Already, the finding that Peruvian rock glaciers are among the world's highest elevationally could be an interesting data point in such comparisons, possibly useful for climate model validation and climate change indicators in high tropical mountains.

From an applied perspective, PRoGI v1.0 is also significant in Peru for natural resource management and hazard mitigation. Rock glaciers store water in frozen form and slowly release it, which can sustain baseflows in arid areas during dry seasons (Jones et al., 2018a). This inventory's maps of rock glacier locations and estimated extents allow hydrologists to include these landforms in water resource assessments, particularly for southern Peru where they may contribute non-trivial to streams dry-season flow (Schaffer et al., 2019).

Additionally, as climate warms, there is concern that destabilization of rock glaciers, through ice melt or permafrost thaw in adjacent steep rock walls, could pose risks in the form of landslides or debris flows. By identifying which rock glaciers are transitional or likely degrading, the inventory dataset helps pinpoint locations that might experience increased geomorphological activity. This information can support decision-making by agencies such as Emergency Operations Centers (COE) at the local, regional, and national levels, as well as public institutions like ANA and INAIGEM, by helping prioritize
specific watersheds for detailed hazard assessments. This work has effectively been provided a country-wide inventory for climate adaptation planning, something that did not exist before.

- Looking forward, several lines of future work are planned based on this inventory: Temporal monitoring: now that this baseline is set, repeat satellite observations (e.g., in 5–10 years) or the analysis of time series (like the 2017 vs 2024 imagery) could reveal if any rock glaciers are retreating at the margins or if new ones are forming.
- Although it is unlikely in the current warming trend, perhaps small protalus features might start creeping. Also, the application of InSAR (Interferometric Synthetic Aperture Radar) data could directly measure surface velocities of rock glaciers. This would allow validation of the active vs. transitional classifications and might identify a subset of "active" rock glaciers with significant movement (>1 cm/year). It is planned to use Sentinel-1 InSAR data as well as high-resolution DEM differencing in a follow-up study. Field validation: fieldwork is essential to truly confirm ice presence. Targeted



field investigations are proposed on a few accessible rock glaciers in each subregion. Techniques such as Ground Penetrating Radar (GPR) or borehole drilling can confirm ice thickness. Field observations can also refine our understanding of surface characteristics (e.g., checking for springs at fronts, vegetation differences) that might distinguish transitional vs. relict in marginal cases. Such ground truth would improve interpretation of remote sensing data.

- Climate/permafrost modelling: with this inventory, models of permafrost distribution in the tropical Andes can be
calibrated. For example, one could run a permafrost model forced by observational data and see if it predicts permafrost where we see rock glaciers. Conversely, using the rock glacier locations as input, one can attempt to reconstruct paleoclimate conditions that would have allowed them to form in currently marginal areas (similar to what Rangecroft et al., 2014 did for Bolivia). This could yield insight into late Holocene climate fluctuations in Peru.

- Expansion to a global inventory: finally, PRoGI v1.0 can contribute to the anticipated global rock glacier inventory that
the IPA action group is working towards. By ensuring our data is formatted according to their specifications, we effectively add the tropical Andes piece to that puzzle. As global datasets (like the one by Sun et al., 2024 for Tibet) become available, comparative studies can assess, for instance, how tropical rock glaciers differ in average size or elevation from similar landforms in other ranges and latitudes.

   Finally, the establishment of this high-resolution dataset is a major step for Andean geoscience and cryosphere research. It
creates a platform for diverse investigations – from basic permafrost science to practical applications in water management. We stress that the inventory is not static: it will be updated as new imagery and techniques arise, ensuring it remains a relevant and accurate resource. By doing so, we aim to keep the scientific community and decision-makers equipped with the latest knowledge on where and how much permafrost exists in the Peruvian Andes, and how it is evolving in our changing climate.

## 7 Data availability

The PRoGI v1.0 dataset is available from the PANGAEA repository at:

- Geopackage: https://doi.pangaea.de/10.1594/PANGAEA.983251 (Medina et al., 2025a).
- CSV file with Geomorphological and topoclimatic characteristics: https://doi.pangaea.de/10.1594/PANGAEA.983295 (Medina et al., 2025b).

## 8 Conclusion

This study presents the first comprehensive inventory of rock glaciers in the Peruvian Andes – the Peruvian Rock Glacier Inventory v1.0 (PRoGI v1.0) – comprising 2338 rock glaciers across a total area of 94.09 ± 0.05 km². Developed following internationally recognized IPA guidelines, PRoGI v1.0 addresses a critical data gap in the tropical Andes and provides a standardized baseline for scientific and environmental applications.

Key contributions and implications:





- Benchmark dataset: PRoGI v1.0 establishes a much-needed benchmark for mountain permafrost in Peru. The inventory reveals distinct spatial patterns: rock glaciers are heavily concentrated in the southern Peruvian Andes (SDOT and SWOT subregions) at elevations of ~4800–5200 m a.s.l., predominantly on south- to southwest-facing slopes. These patterns reflect the sensitivity of periglacial features to the interplay of local climate (temperature and precipitation) and topography. The absence or scarcity of rock glaciers in the northern Andes highlights the climatic conditions necessary for their existence. This dataset thus provides insight into the current extent of permafrost-related landforms under ongoing climate conditions.

- Global integration: By adhering to the standard protocols of the IPA Action Group, our inventory is fully interoperable with rock glacier inventories from other regions. This interoperability means Peruvian data can be directly compared or merged with data from, e.g., the European Alps or Himalayas. As a result, PRoGI v1.0 contributes to broader efforts like the Rock Glacier Inventory and Kinematics (RGIK) initiative. It enables researchers to include tropical Andean permafrost in comparative studies, improving our understanding of how rock glacier characteristics vary across different mountain systems and climatic zones.

- Environmental management: The high-resolution and nationwide scope of the inventory has practical implications. For water resource management, PRoGI v1.0 pinpoints where hidden ice reserves exist in the Andes (outside of glaciers). This information can feed into hydrological models, especially in arid regions of southern Peru where rock glacier meltwater may sustain dry-season streamflow. For hazard assessment, the inventory identifies numerous steeps, ice-cored landforms; as climate warms, some of these could destabilize under continued warming, thus warranting monitoring. Government agencies (e.g., ANA, CENEPRED) can use this publicly available dataset as a decision-support tool to prioritize field investigations or adaptation measures in high mountain areas sensitive to permafrost changes.

While PRoGI v1.0 is a major step forward, we note that it represents a snapshot in time of rock glacier distribution. Given the continued climate warming, the state of some rock glaciers will evolve – for example, currently transitional rock glaciers may become relict in the coming decades. Therefore, we envisage the inventory as dynamic: it will be updated as new data and techniques become available. Future improvements will include the integration of InSAR-based kinematics to directly measure rock glacier movement, which will refine the activity classifications by providing independent velocity data (RGIK, 2023).

We also plan to incorporate next-generation high-resolution optical datasets (e.g., Planet or forthcoming sub-meter satellites) to detect any smaller landforms and to improve boundary precision. Furthermore, periodic updates (e.g., PRoGI v2.0 in a few years) will document changes in rock glacier extent or activity, thereby serving as a monitoring tool for the effects of climate change on Peru's cryosphere. Field campaigns will complement these efforts by validating the presence of ice and monitoring ground temperatures at selected sites.

In conclusion, the PRoGI v1.0 dataset provides an essential foundation for advancing our understanding of permafrost in the tropical Andes. It delivers both the big-picture view (national distribution patterns) and the fine details (individual landform characteristics) needed for cross-disciplinary research. We expect that this inventory will spur new research on Andean

permafrost dynamics, contribute valuable data to global assessments of mountain cryosphere, and inform local strategies for water resource management and hazard mitigation in a warming world.

**Author contributions**

**KM** (Katy Medina) led the study and contributed to conceptualization, data curation, methodology design, visualization, and wrote the manuscript draft. **HL** (Hairo León) co-led conceptualization performed data curation and GIS analyses and co-wrote the original draft. **EL** (Edwin Loarte) contributed to study design, supervised the project, and participated in writing – review and editing. **EBR** (Edwin Badillo-Rivera) contributed to data curation (especially field data integration) and methodology and assisted with writing – review and editing. **XB** (Xavier Bodín) advised on methodology (especially regarding IPA standards) and supervision and contributed to writing – review and editing. **JU** (José Úbeda) provided expertise in analysis and interpretation, supervised aspects of the research, and contributed to writing – review and editing.

**Competing interests**

The authors declare that they have no competing interests.

**Acknowledgements**

We thank all colleagues and institutions that supported this work. Sebastián Vivero, Guillermo Azócar, Christian Huggel, and Lidia Ferri are especially thanked for their valuable comments and suggestions during the development of the inventory. The research team is also grateful to Hildebrant Flores, Ricardo Valverde, and Alex Mendoza for their collaboration in field data collection and insights into local periglacial conditions. We also appreciate the constructive feedback from the 16 external reviewers (listed in Table S1) who rigorously evaluated the draft inventory; their input greatly improved the quality of the final dataset.

**Financial support**

This research was supported by the project "Permafrost: rock walls and rock glaciers as water regulators and risk generators in Peru in a context of climate change (PermaPeru)", which is funded by CONCYTEC-PROCIENCIA (Contract N° PE501089264-2024-PROCIENCIA). Additional support was provided by the Center for Research in Earth, Environmental and Climate Sciences and Technologies (ESAT) of the Universidad Nacional Santiago Antúnez de Mayolo (UNASAM), through its Mountain Ecosystems and Permafrost Research Group (PAMEC).



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
