# Peer review of "A high-resolution dataset of Rock Glaciers in the Peruvian Andes (PRoGI): inventory, characterization and topoclimatic attributes."

_Earth System Science Data, 2025_

## Author Comment (AC2)

**Answers to Referee #2**

Below, the reviewer's comments appear in blue and our responses in **black bold**. References in the manuscript appear in *italics*. Changes made to the manuscript (revised version) are underlined. Line numbers refer to the original preprint.

The authors present a geomorphological rock glacier inventory of the Peruvian Andes (i.e., PRoGI) compiled through the manual mapping on Bing (and Google Earth) optical imagery. My comments are mainly concerned with the manuscript. I didn't have time to look at the actual shapefile. In my view, the paper is suitable for publication in ESSD after substantial revisions.

I suggest improving the logical flow of the introduction/results/discussion and provide stronger (basic and applied) motivations for compiling the inventory in the way it is presented/discussed in this paper. Considering that RG degree of activity in PRoGI relies on the visual assessment/interpretation of landforms, I suggest reducing substantially the length of the analysis (and accompanying text of the Results and the Discussion) concerned with the classification of rock glaciers into active, transitional, and relict. My suggestion is motivated by the high degree of uncertainty that is typically associated with the morphologically based classification of rock glacier activity. Please see my detailed comment to section 5.2.1. Overall, I believe the paper would benefit from substantial trimming. A more concise paper structure would allow the main original points of the inventory to stand out more apparently.

Please consider all of my comments on the constructive side. Thank you for your effort on this work.

Thank you very much for your positive assessment of our research and for your valuable comments on how to improve this manuscript.

**General comments - manuscript:**

1. TITLE: On the "high-resolution" character of the inventory I share the evaluation made by referee #1. Nowadays, the use of Bing and GE imagery (comprised between 1 and 5 m resolution) represents the norm. A high-resolution inventory, in my view, should employ sub-metric RGB imagery, and possibly LiDAR-derived DTMs for increasing three-dimensional perspective and filtering out vegetation cover over vegetated RGs. For example, this is the case of a geomorphological inventory recently compiled across South Tyrol (Italy) by Scotti et al. (2024), who utilized 0.2-to-0.5 m gridded orthophoto mosaics and a 2.5 m LiDAR DTM. Incidentally, this inventory tallies a number of rock glaciers (n = 2798) comparable to PRoGI, and may serve as a useful term of comparison in Table 8.

The difference between an inventory compiled on GE imagery and one compiled on higher-resolution orthophoto mosaics coupled with LiDAR-derived DTMs was

assessed by Brardinoni et al. (2019). Accordingly, it was found that "the number of mapped rock glaciers on GE imagery exhibited higher inter-operator heterogeneity (up to a factor of 3), and that using LiDAR and higher resolution orthophotos lowers this heterogeneity down to a factor of 2, while producing an increase in the number of mapped landforms, which become systematically smaller".

To summarize, I believe that image resolution matters and the resolution of Bing and GE imagery would not warrant a consistent "high resolution" inventorying output across a large study area such as the Peruvian Andes.

Thanks for the comment. We recognize that the term "high-resolution" can be subjective and that Bing/Google Earth images (1-5 m) are standard for the development of regional inventories. We have removed "high-resolution" from the title and manuscript, focusing on the main contributions of PRoGI as the first comprehensive inventory covering the entire Peruvian Andes and providing detailed topoclimatic attributes using standardized mapping protocols. The proposed new title is: "A comprehensive rock glacier inventory for the Peruvian Andes (PRoGI): dataset, characterization, and topoclimatic attributes."

However, we have retained the term to describe images from Google Earth and Bing Maps, as other studies recognize that these satellite images belong to the high-resolution category for remote sensing data (Abdullah and Romshoo, 2024; Bhat et al., 2025).

Abdullah, T., & Romshoo, S. A. (2024). A Comprehensive Inventory, Characterization, and Analysis of Rock Glaciers in the Jhelum Basin, Kashmir Himalaya, Using High-Resolution Google Earth Data. Water, 16(16), 2327.

Bhat, I. A., Rashid, I., Ramsankaran, R. A. A. J., Banerjee, A., & Vijay, S. (2025). Inventorying rock glaciers in the Western Himalaya, India, and assessing their hydrological significance. Geomorphology, 471, 109514.

 STUDY AREA: The authors adopt a climatic classification of the Peruvian Andes proposed by Bonshoms et al. (2020), which in turn is based on previous climatic characterization of glaciers across the entire South America (Sagredo and Lowell, 2012).

Sagredo, E.A. and Lowell, T.V. (2012) Climatology of Andean glaciers: a framework to understand glacier response to climate change. Global and Planetary Change., 86-87, 101–109.

In the results, and especially in the discussion, it will be important to remind the reader that the four broad regions were defined solely based on climate, and therefore they do not consider interactions with the relevant terrain altitudinal distribution (i.e., how much area is available in each region for RG development

above (current and former) critical isothermal altitudinal thresholds) and the dominant lithologies (i.e., propensity for rock walls to disintegrate in blocky debris, hence promote thermal ventilation and permafrost persistence). Having considered climatic regions only may explain the lack of explanatory power and the intra-regional heterogeneity observed in terms of RG spatial distribution. In this context, the subdivision of South Tyrol in physiographic zones (i.e., combining broad climatic, hypsometric, and lithologic characteristics) offered an obvious advantage, in explaining the relevant spatial variability in rock glacier density (Scotti et al., 2024).

A description (even a brief one would suffice) of the geological setting, including a list of the dominant lithologies in each of the four climatic regions is missing. Please consider adding one.

We thank the reviewer for this valuable suggestion. We have expanded the Study Area section to include geological context based on our analysis of lithological distribution across the four climatic regions. Specifically, we have added:

- 1. Quantitative geological description: Added a detailed breakdown of dominant lithologies in each climatic subregion, including the main geological formations:
  - "...From a geological point of view, the four subregions have distinct lithological characteristics that influence the potential for rock glacier formation. The SDOT subregion is dominated by Miocene-Neogene volcanic-sedimentary sequences (Nm-vs), which produce abundant block debris for rock glacier development. The NWOT subregion is mainly characterized by Neoproterozoic schists and gneisses (NP-esq,gn), which break down into competent debris. The NDOT contains important Paleogene-Neogene volcanic and sedimentary formations (PN-vs), while the SWOT mainly sits on Ordovician metasedimentary rocks (O-ms)."
- 2. Hypsometric context: The results include an analysis of the altitudinal distribution of rock glaciers that may be comparable to critical permafrost thresholds.
- 3. Explicit clarification: we have clearly indicated that the regional classification is based solely on climate:
  - "It is important to emphasize that this regional framework is based solely on climatic criteria and does not incorporate other factors"
- 3. METHODOLOGY: In the present inventory the authors partly follow the RGIK guidelines, partly do not. Consequently, the inventory would not be directly comparable with other RGIK-based counterparts compiled elsewhere around the globe. I am not suggesting that the inventory should adopt the entire RGIK identification, location, characterization, and delineation protocol; however, it is

important that the authors: (i) adhere to the mandatory parts of the RGIK guidelines; and (ii) clearly summarize which components of the RGIK methodological workflow are adopted, which not, and whether a different nomenclature is applied for some of the attributes.

For example, RGIK considers as mandatory component the compilation of a shapefile of primary markers. That is, a shapefile made of point elements reporting lat, long and unique identifier for each rock glacier unit (RGU) and system (RGS). Following this logic, each rock glacier is hierarchically classified into units and systems. Similarly, the RGIK morphological-based activity classification does entail classifying rock glaciers units into active, active uncertain, transitional, transitional uncertain, relict, and relict uncertain. The present inventory encompasses solely active, transitional, and relict landforms. This is a potentially critical approach, considering the inherent uncertainty associated with the classification of transitional rock glaciers i.e., note the high degree of discrepancy between international experts in the activity classification of inactive/transitional landforms (Brardinoni et al, 2019). Indeed, the test presented by Brardinoni et al involved not only RG outline delineation (see section 4.6 of the present manuscript), but also the activity classification.

We thank the reviewer for this important observation regarding RGIK guideline implementation. We have substantially revised the Methodology section to explicitly address compliance:

- Implementation of primary markers: We have created and attached a complementary geopackage file of primary markers (in the PANGAEA repository) with hierarchical RGU/RGS classification, thus complying with this mandatory RGIK requirement.
- 2. Explicit statement of compliance: A brief summary of the RGIK guidelines that have been adopted in full, in part, or modified has been added, along with a justification for each decision.
  - "..., with specific adaptations for the scale and context of our national inventory.:
  - (1) In accordance with the mandatory requirements of RGIK, we compiled two vectors (in \*. gpkg format), the first containing the primary markers for each rock glacier unit or system and the second, the extended footprint of the polygonal delineation for each rock glacier including its associated attributes.
  - (2) In the activity classification approach, while RGIK recommends six activity classes (including "uncertain" categories), we employed a simplified three-class system (active, transitional, and relict) due to the extensive spatial coverage required for a practically implemented system.

- (3) The scheme for the inventory attributes has been partially adopted from the RGIK guidelines, as they have been complemented with topoclimatic variables for hydrological and climatic applications."
- Justification of activity classification: We provide a comprehensive explanation of our simplified three-class system, acknowledging the uncertainty inherent in transitional forms but justifying our approach based on consensus mapping and the practical constraints of nationalscale inventorying.

**4. RESULTS**

Please consider whether all of the sub-sections are needed to convey the main message of the paper. Some may be deleted; some may be merged. I feel that some descriptive information does not lead to original insights. Currently, the results are subdivided in many sub-sections, many of which consist of a single paragraph.

This paper section (the Results) contains both results and interpretations. Please move interpretations to the Discussion e.g., lines 323-325 and lines 332-333. Just to mention a couple of examples that I have noticed in subsection 5.1.

We sincerely thank you for your thorough review and your valuable comment, which have significantly helped us to improve the manuscript's clarity and focus. We have carefully considered your feedback regarding the structure of the Results section and the mixing of results with interpretation.

We agree with the first observation and have thoroughly revised the structure of the Results section to reduce fragmentation and improve the narrative flow. Specifically, we have merged several sub-sections:

- The original subsections 5.3.1 "Slope distribution" and 5.3.2 "Aspect (orientation)" have been merged into the main 5.3 "Topographic and Climatic Attributes".
- The subsection 5.2.1 "Rock glacier activity" has been integrated into 5.2 "Rock Glacier Characteristics: Morphology and Activity", creating a more cohesive presentation of the landforms' properties.
- This consolidation has reduced the total number of subsections and eliminated those comprising only a single paragraph, resulting in a more streamlined and logically structured section.

We have meticulously reviewed the entire Results section and have removed all interpretive statements, relocating them to the Discussion section. This includes, but is not limited to, the examples you kindly pointed out:

 Interpretations regarding the influence of climate aridity, lithology, and local geomorphological factors on rock glacier distribution (from Section 5.1) have been moved.

- Explanations about the significance of morphological shapes and the implications of the activity status statistics (from Section 5.2) have been transferred.
- Discussions on why certain slope angles or southerly aspects are preferential (from Section 5.3) have been relocated.

**5. DISCUSSION**

Based on the revised structure of the Results, please consider which parts of the Discussion may be really necessary to highlight the originality and robustness of your inventory, and which others may be just chocking the reader with unnecessary details (or debatable interpretations). I believe that the overall reading of the paper would highly benefit from some Discussion simplification.

Based on the above and the need for substantial revision, I will limit my comments on the Discussion to Table 8. Considering that this manuscript is not a review paper, please consider restricting your list to inventories that can help making a more straightforward and meaningful comparison with PRoGI. For example, wouldn't be enough comparing PRoGI to other inventories in South America? Even when a worldwide comparison was deemed necessary, what is the point of comparing PRoGI with inventories that contain less (or a little more) than 100 RGs? To warrant a more reliable comparison, I would limit Table 8 to inventories that encompass thousands (or at least several hundreds) of RGs.

We sincerely thank the referee for this overarching suggestion, which has significantly improved the clarity and impact of our Discussion. We have undertaken a major simplification and restructuring:

- We merged and condensed paragraphs to avoid repetition.
- We eliminated contentious interpretations and overly detailed comparisons that did not directly highlight the main findings.

We fully agree with this logic. Following the referee's recommendation, we have substantially revised Table 8 to include only inventories that allow for meaningful and direct comparison. We have applied clear criteria, restricting the table to inventories containing several hundred rock glaciers or more, and we have prioritized major inventories from South America and other key global regions that serve as essential reference points. This makes the table more concise, reliable, and directly relevant to the scale of our study.

**Specific comments:**

Line 39: "Nowhere is this more evident ...". Please consider tuning this sentence down. The pace of warming is widely documented in mountain and high latitude area around the globe, and the tropical Andes are just one of them.

We thank the reviewer for this suggestion. We have modified the sentence to tone down the absolute claim, replacing "Nowhere is this more evident than in the tropical Andes..." with "This warming trend is particularly evident in the tropical Andes..." to acknowledge that other mountain regions are also experiencing significant warming while still emphasizing the relevance of our study area.

Line 43: unclear in what sense RGs would stand out as "geomorphological archives". Please expand and add references testifying to the geomorphic archival value of RGs.

We have expanded this concept to clarify what we mean by "geomorphological archives" and added appropriate references: "Among periglacial landforms, rock glaciers stand out as both geomorphological archives of past climate conditions, preserving information about paleo-temperatures through their internal structure and development history (Haeberli et al., 1999) and as vital water reservoirs."

Line 43-45: if adopting the RGIK (2023) technical definition of rock glaciers, please report the complete version (i.e., creep and "shearing at depth"; "optionally" ridge ad furrow topography, since these features do not always occur). Presently, the definition provided would apply just to active landforms (i.e., steep fronts).

We have updated the definition to include the complete RGIK (2023) version: "These ice-debris landforms, formed by the creep of ice-rich permafrost and shearing at depth, optionally exhibit steep fronts, lateral margins, and ridge-and-furrow surface topography (RGIK, 2023).

Line 45: Boccali et al (2019) refers to the southeastern European Alps (the Julian Alps), which are not exactly an arid region, similar to the Peruvian Andes. Consider replacing this reference with a better fit or rewriting the sentence.

We have replaced the inappropriate reference with more relevant Andean studies: "Comprising 15–70 % ice by volume (Halla et al., 2021; Haq and Baral, 2019), rock glaciers store substantial water equivalents in arid regions like the southern Peruvian Andes (Janke et al., 2017; Rangecroft et al., 2015; Schaffer et al., 2019)."

Lines 46-47: The thermal insulation (through ventilation) afforded by the surficial blocky layer is a property of rock glaciers that has been known/characterized for decades. Besides Brighenti et al (2021), please add reference to prior studies that have indeed characterized with empirical data the internal structure of rock glaciers e.g., Scapozza et al (2011), Geomorphology.

We have added the suggested reference and other key studies: "Their debris mantle confers thermal inertia through ventilation effects, buffering ground ice against short-term climate variability (Brighenti et al., 2021; Scapozza et al., 2011)."

Lines 46-47: "This dual role as climate sentinels and hydrological buffers makes rock glaciers indispensable for understanding long-term environmental change". Please clarify what is meant by climate sentinels. Are they considered climate sentinels across the Quaternary or at shorter contemporary time scales? In the former case, please add reference to Quaternary studies involving numerical dating of RGs (e.g., 10Be, 14C). In the latter case please add reference to Rock Glacier Velocity as an emergent variable of climate change.

We have clarified both temporal scales: "This dual role as climate sentinels—providing insights into both contemporary climate change through velocity monitoring (Kääb et al., 2021) and Quaternary climate history through dating of their formation (Palacios et al., 2020)—and hydrological buffers makes rock glaciers indispensable for understanding environmental change across multiple timescales.

Line 48: please cite a reference when stating the definition of permafrost.

We have added the requested reference: "Mountain permafrost, defined as ground remaining ≤0 °C for at least two consecutive years (van Everdingen, 1998), underpins these systems."

Line 54: To acknowledge alternative views on rock glacier origin/formation that are still matter of international debate (i.e., permafrost vs glacier-to-rock glacier transition), please consider adding a sentence in which you state that in this paper you do not address the question of ice origin and rock glacier formation.

We have added the suggested clarification: "It should be noted that while the origin of rock glaciers (permafrost creep vs. glacier-to-rock glacier transition) remains debated internationally, this paper focuses on their morphological characterization and distribution without addressing formation mechanisms."

Line 60 "the first high resolution" and Line 63 "By combining sub-meter remote sensing imagery". Given the imagery utilized (GE and Bing imagery ranging between 1 and 5 m cell size, as reported in Table 1, are not sub-metric), I don't see the point of considering the inventory a high-resolution one. This comment applies to the paper title too.

We have removed "high-resolution" from the description of PRoGI and corrected the resolution specification: "To address this, we present the Peruvian Rock Glacier Inventory (PRoGI v1.0), the first nationally comprehensive rock glacier dataset for the Peruvian Andes, compiled using the mapping standards of the International Permafrost Association's Action Group (RGIK, 2023). By combining high-resolution remote sensing imagery (0.5-5 m) with rigorous geospatial analysis,

PRoGI v1.0 documents the distribution, morphology and climatic characteristics of rock glaciers across Peru."

Lines 69-70: for the sake of conciseness, please consider removing these two lines. I think the introduction would stand up fine without them.

We have accepted your suggestion to remove those two lines, thank you very much.

Line 118: "independent check". Please expand on the likely reliability of the modelled MAGT. Similar models are known to work relatively well in low-lying Artic regions but subject to large uncertainty in rugged mountain terrain, due to high spatial heterogeneity in sediment texture and other local variables. Most importantly, how did you proceed when a RG was labelled relict but plotted withing MAGT

Figure 2. Rock glacier mapping examples: (a) Classical case with well-defined morphology (15°37'3.79" S, 72°23'0.32" W) showing clear frontal and lateral margins; Complex case with mapping challenges (17°04'30" S, 69°59'24" W) showing truncated frontal morphology and uncertain lateral boundaries.

4. Transparent discussion of mapping challenges: We now explicitly describe the complexities and uncertainties in delineating challenging rock glaciers like the one originally shown, explaining our conservative approach in such cases.

These modifications provide greater transparency about our methodological choices and mapping challenges.

**4.1.3 Quality Control**

In the RGIK protocol, the identification of RGs comes as a first step, which involves mapping primary markers and then classifying them as: rock glaciers, uncertain rock glaciers, and not rock glaciers. This means that an RGIK inventory ultimately retains some primary markers labelled as "uncertain rock glaciers".

In the case of PRoGI, please clarify how uncertain landforms were dealt with. I can imagine that a subset of landforms has remained uncertain. Were these landforms

excluded or retained in the inventory? The last two sentences (lines 187-189) are not clear on this point.

Line 171: please replace the full stop with colons at the end of the sentence and before the list.

Line 172: Sun et al 2024 is most likely a repetition and should probably be deleted from the first bullet point.

We thank the reviewer for these important clarifications. We have revised this section to address all points:

- 1. Uncertain landforms clarification: We explicitly state that uncertain features were either resolved through consensus review or excluded from the final inventory if identification remained ambiguous.
- 2. Colon correction: We have replaced the full stop with colons as suggested.
- 3. Repetition removal: We have deleted the redundant "Sun et al., 2024" reference from the first bullet point.
- 4. RGIK alignment: We clarify how our certainty levels (0 and 1) correspond to RGIK's "uncertain rock glacier" category and describe our resolution process.
- 4.2 Geomorphological identification criteria

Based on its content, this part does not deal with the identification but with the classification of the activity status (or degree of activity). Please change the title of this sub-section accordingly.

Some of the morphological and thematic criteria utilized here hold the risk of reading too much in terms of activity, while not disposing of reliable kinematic data. For example:

1) Vegetation: I don't find particularly useful using vegetation as a possible criterion for discriminating relict from active and transitional counterparts. Borrowing criteria and evidence drawn from wetter physiographic regions, such as the European Alps (e.g., Scotti et al, Colucci et al, Kellerer-Pirklbauer) is not particularly reliable in arid regions of the Peruvian Andes, where hardly any vegetation can grow in similar dry settings.

This is explicitly stated in page 10 of RGIK (2023): "In arid regions, vegetation may nevertheless be lacking on relict rock glaciers due to unfavorable environmental conditions".

2) Ridge and furrows: this morphological element is not considered in the RGIK guidelines as diagnostic evidence for discriminating between active, transitional, and relict rock glaciers.

Based on the above, I wonder to what extent the morphological criteria detailed in Table 2 would lead to the compilation of an inventory consistent with existing (or forthcoming) RGIK-based inventories conducted elsewhere on Earth. Please elaborate.

We thank the reviewer for these important observations about activity classification criteria. We have implemented the following changes:

- 1. Title modification: Changed to "4.2 Geomorphological identification and classification" to accurately reflect the content.
- 2. Vegetation criterion clarification: We have added a qualification noting that vegetation is a secondary indicator with limited applicability in arid regions, citing the RGIK (2023) caution about arid environments.
- 3. Ridge and furrows context: We clarify that while ridge and furrow topography is optional in RGIK guidelines, it has been used as supporting evidence in multiple Andean studies, though not as a primary diagnostic criterion.
- 4. Criteria prioritization: We now emphasize that our classification relied primarily on frontal morphology and overall landform integrity, with other criteria serving as supplementary indicators.

**4.4 Classification of rock glaciers**

The first part of this section (lines 250-266) largely duplicates what described in section 4.2. Please move these 16 lines of text in section 4.2, ensuring to avoid repetitions.

The second part of this section is not entirely convincing, since it is mixing up the geometric characterization of simple/monomorphic landforms (termed units according to RGIK 2023) into lobate and tongue-shaped morphologies, with those of multilobe/polymorphic morphologies (termed systems according to RGIK 2023). Consequently, this hybrid system of classification does not deal solely with geometry, but also with the ability to distinguish (or not) different rock glacier units (in terms of front, lateral margins, and debris source) within a system, due to adjacency, coalescence, and overlapping of lobes.

The present classification scheme does not offer a clear protocol for consistently discriminating between complex and simple rock glacier configurations. In this context, the main rationale for proposing an RGIK hierarchical classification of rock glaciers into units and systems was exactly to mitigate mapping heterogeneity among operators when dealing with complex (multilobe) morphologies.

Please inform the reader about which geomorphic insights on rock glaciers may be gained by implementing the geometric characterization described in lines 273-280 and Table 5, and in particular the distinction between lobate and tongue-shaped morphologies. After reading section 5.2, which deals with the descriptive statistics

on the above RG classification, I could not find any geomorphic insight that could help better understanding rock glacier occurrence (and relevant environmental controls) in the landscape.

We thank the reviewer for this important clarification about geometric classification. We have substantially revised this section to:

- Align with RGIK hierarchical framework: We explicitly connect our geometric classification to the RGIK unit/system hierarchy, clarifying that lobate and tongue-shaped forms represent Rock Glacier Units (RGUs), while coalescent and polymorphic forms represent Rock Glacier Systems (RGS).
- 2. Clarify classification protocol: We describe the specific criteria used to distinguish between simple units and complex systems based on debris source differentiation and lateral margin discernibility.
- 3. Justify geomorphic relevance: We explain how geometric classification provides insights into debris supply mechanisms, topographic constraints, and developmental history of rock glaciers in the Andean environment.
- 4. Connect to primary markers: We reference our implementation of RGIK primary markers that document this hierarchical classification.

**"4.4 Geometric classification**

Our geometric classification aligns with the RGIK hierarchical framework, distinguishing between Rock Glacier Units (RGUs) representing individual landforms and Rock Glacier Systems (RGS) comprising multiple coalesced units. This approach addresses mapping consistency for complex morphologies while providing insights into debris supply patterns and topographic controls.

For simple/monomorphic landforms (RGUs), we classified geometry based on planform characteristics:

- Tongue-shaped: length-to-width ratio >1, indicating downslope-oriented flow (Harrison et al., 2008; Humlum, 1982).
- Lobate: length-to-width ratio <1, characteristic of cirque-floor or slope-base accumulation (Humlum, 1982).

For complex configurations (RGS), we identified:

- Coalescent: composite features formed by convergence of multiple tongueshaped lobes with discernible individual sources (Humlum, 1982).
- Polymorphic: heterogeneous assemblages displaying multiple geometric forms within a single system, often indicating complex developmental histories (Falaschi et al., 2015).

The distinction between units and systems was based on the discernibility of individual frontal and lateral margins, debris source differentiation, and spatial separation of constituent lobes. This geometric classification, documented through our RGIK-compliant primary markers, provides insights into:

- Debris supply mechanisms and source area characteristics
- Topographic constraints on rock glacier development.
- Spatial organization of periglacial processes in different Andean environments."

**4.6 Uncertainty assessment**

Besides evaluating between-operator heterogeneity in terms of polygon delineation, how was the uncertainty assessment conducted in terms of degree of activity?

We thank the reviewer for this important question about activity classification uncertainty. We have expanded this section to explicitly describe how we assessed and addressed uncertainty in activity classification through our consensus mapping protocol and multi-analyst validation process.

**Subsection 5.1**

Table 5: please consider moving this table in the supplementary file, while retaining just one line for the summary data relevant to each of the four climatic regions. Currently, the table contains basin-specific information which appear excessive, considering that the reader is not provided with any significant information on such drainage basins. For evaluating the relative spatial distribution of rock glaciers (abundance/paucity), I recommend that the authors use the number of rock glaciers per unit terrain area (rock glacier density) – as opposed to simple rock glacier count. Rock glacier density will be directly comparable across the relevant climatic regions. Presently, the reader does not know how large the different climatic regions are. In the text, the authors mention rock glacier density a few times, but no systematic analysis/evaluation is shown.

We thank the referee for these excellent and constructive suggestions. We have implemented both changes, which have significantly improved our spatial analysis.

- Moving Table 5: As recommended, we have moved the detailed basinspecific Table 5 to the Supplementary Material (now Table S2).
- 2. Creating a new summary table for the main text (new Table 5). To enhance clarity and comparability, we present the density as the number of rock glaciers per 100 km2 and the area coverage as a percentage (%).

Table 1. Count and density of rock glaciers by subregion.

| Subregion | Count | Total area (km²) | Subregion area (km²) | Density
(per 100 km²) | Area coverage (%)
(per 100 km²) |
|-----------|-------|------------------|----------------------|--------------------------|------------------------------------|
| NDOT      | 17    | 0.63             | 16 061.03            | 0.106                    | 0.004                              |
| NWOT      | 47    | 1.89             | 91 321.86            | 0.051                    | 0.002                              |
| SDOT      | 2135  | 87.73            | 107 643.97           | 1.983                    | 0.082                              |
| SWOT      | 139   | 3.84             | 93 097.51            | 0.149                    | 0.004                              |

This new analysis provides a systematic and quantitative evaluation, clearly demonstrating that the Southern Dry Outer Tropics (SDOT) host a rock glacier density nearly 20 times higher than the other subregions. The text in Section 5.1 has been revised to discuss these findings.

Lines 322-325: "Lithology is also likely a key factor. The highest percentage of inventoried rock glaciers coincides with volcanic rock outcrops, where the type of chemical alteration increases the albedo of the surfaces and enhances permafrost development and preservation (Yoshikawa et al., 2020)". Please consider expanding on lithology or completely neglecting this factor in the manuscript. A sentence that relates RG abundance on chemical weathering (due to albedo) of volcanic rocks appears too much of a stretch. Lithological effects on rock glacier abundance have been difficult to isolate in the literature. I wouldn't try to solve or dismiss such a complicated topic with a similar sentence on volcanics, which by the way encompass a range of lithological types. For a brief discussion on the complexity of isolating lithological effects (due to spurious interactions with hypsometry and climate), the authors may have a look at Section 4.2 in Scotti et al (2024).

We thank the referee for this critical observation and for pointing us to the highly relevant discussion in Scotti et al. (2024). We agree that the original sentence in the Results section was an oversimplification of a complex factor. In accordance with the referee's suggestion and our own goal to separate results from interpretation, we have removed this sentence from Section 5.1 (Results). A brief discussion of the influence of lithology on the distribution and preservation of rock glaciers was included in the Discussion section, acknowledging the difficulties in isolating lithological effects from hypsometric and climatic controls, and citing both Yoshikawa et al. (2020) and Scotti et al. (2024).

Lines 332-333: "This suggests that even basins in close proximity, under the same broad climatic subregion, can exhibit different rock glacier densities and size distributions – possibly due to local geomorphological factors (such as basin lithology or glacial evolution)". As per prior comment, this sentence appears vague without adding significant information to the paper. Moreover, RG densities are

mentioned, but I couldn't see any quantitative data on this variable. Please consider deleting this sentence.

Thank you very much for your comment. Following the instructions for restructuring this subsection, we have removed and added new text to better define the study's findings and avoid writing assumptions.

Table 6: 1) what was the rationale for the selection of the size bins shown in this table? Two categories contain respectively just 3 and 1 rock glaciers only. The current binning does not seem appropriate. 2) What is the underlying hypothesis for presenting RG size categories as a function of RG mean elevation and slope? I don't understand why slope and elevation should change systematically with RG size, neither I recall any empirical relation constrained along these lines in other inventories. Please consider deleting from Table 6 columns reporting mean elevation and mean slope.

We thank the referee for these insightful comments. We agree that the size bins in the original Table 6 were suboptimal and that the inclusion of mean elevation and slope within those arbitrary categories was not well-justified.

In response, we have removed Table 6 entirely. Instead, we have created a new Figure 5 (provided below) that provides a more integrated and spatially explicit overview of key rock glacier attributes. This new figure includes maps of rock glacier distribution colored by:

- (a) Total count
- (b) Mean area
- (c) Mean Minimum Altitude of the Front (MAF)
- (d) Mean slope

This approach avoids the problematic and arbitrary size-class binning and allows the reader to visually assess the spatial patterns of these characteristics without implying a direct or systematic functional relationship between them. We believe this is a more robust and informative way to present the data.

**Subsection 5.2.1:**

Considering that RG degree of activity in PRoGI relies on the visual assessment/interpretation of landforms, I suggest reducing substantially the length of the analysis (and accompanying text of the Results and the Discussion) concerned with the classification of rock glaciers into active, transitional, and relict. My suggestion is motivated by the high degree of uncertainty that is typically associated with the morphologically based classification of inactive/transitional rock glacier activity (e.g., Table 3 and Figure 11 in Brardinoni et al., 2019), and the rate of reclassification rock glaciers may undergo, once InSAR kinematic data are integrated (e.g., Table 4 and Figure 11b in Bertone et al., 2024).

Bertone A, Jones N, Mair V, Scotti R, Strozzi T, Brardinoni F. 2024. A climate-driven, altitudinal transition in rock glacier dynamics detected through integration of geomorphologic mapping and InSAR-based kinematic information. The Cryosphere, 18, 2335–2356, https://doi.org/10.5194/tc-18-2335-2024

Indeed, Bertone et al found that 15% of the rock glaciers western South Tyrol were reclassified from relict to intact (or vice versa), as a result of InSAR data integration.

This reclassification rate is likely to increase even more when 3 activity classes (as opposed to just the intact and relict ones) are considered.

We sincerely thank the referee for this critical observation and for bringing the highly relevant studies of Brardinoni et al. (2019) and Bertone et al. (2024) to our attention. We fully acknowledge the significant uncertainty inherent in morphology-based activity classification, especially for distinguishing transitional and relict forms.

In direct response to this comment, we have substantially revised the respective subsection in the Results (5.2 Rock glacier characteristics: activity and morphology).

Line 362: "inactive". Please replace or remove the term inactive, as it would confound the reader. Traditionally, inactive and relict have been used to differentiate two distinct classes of activity.

**We have removed the term inactive from the text.**

Lines 369-372: "Indeed, we found that relict rock glaciers have the smallest mean size (mean area  $\sim 0.03 \text{ km}^2$ ), compared to  $\sim 0.04 \text{ km}^2$  for transitional and  $\sim 0.06 \text{ km}^2$  for active rock glaciers. This pattern is consistent with expectations: once a rock glacier loses its ice (becoming relict), it may slump and shrink over time, whereas active ones are buttressed by internal ice and can maintain larger extents."

Technically, the above sentences belong to the Discussion, as they contain interpretation of the results. Most importantly, please consider rethinking your possible explanation, since rock glaciers are complex landforms associated with millennial time scales of development. Indeed, RG size has to do with age of formation, length of (continuous or discontinuous) activity through millennia, rate of sediment supply, and available room within a valley/slope for growing in planimetric size. Attributing the smaller size of relict rock glaciers to slumping (or other mass wasting styles of obliteration) appears simplistic, considering that the vast majority of rock glaciers degrade (non-catastrophically) through subsidence. Incidentally, this interpretation contrasts with prior results from the Italian Central Alps (Scotti et al 2013), where relict RGs were found to be systematically larger than intact counterparts (i.e., Figure 8, and Figure 6a, cf. gray (intact) and white (relict) box whiskers).

Thank you very much for your comment, and we agree with you. We've removed this sentence and placed a brief discussion in the next section. However, recognizing that long and complex processes can determine the size of rock glaciers and associate them with their activity, we've tried not to be so emphatic in that regard.

Lines 410-412: "This suggests that slopes around 15–20° are especially conducive to rock glacier formation/preservation, likely because they are steep enough for debris-ice creep but not so steep as to cause frequent avalanching or debris removal."

Technically, the above sentence belongs to the Discussion, as it provides an interpretation of the results.

**We have moved the prayer to the discussion section.**

**Subsection 5.3.2**

Figure 8: from the description of how mean RG aspect was computed in QGIS, I suspect that this variable might be biased. The main issue when calculating the average aspect of a surface is that aspect is a circular variable, meaning that the mean is incorrect due to the discontinuity that occurs around 360 degrees for northerly aspects (i.e., aspects approaching single-digit degrees are adjacent to aspects approaching 360 degrees). Typically, this results in underestimating the number of RGs that are dominantly facing north. Please double-check whether adequate transformation was conducted during aspect calculation and elaborate on this in the methods.

We have verified the aspect calculation in our inventory and it has been estimated correctly. Thanks to your suggestion, we have included the description of how it was done in the methods section:

"The mean aspect was calculated using a circular mean transformation (e.g., calculating the mean of sine and cosine components) rather than a simple arithmetic average."

---

## Author Comment (AC3)

**Answers to Referee #1**

Below, the reviewer's comments appear in blue and our responses in black and **bold**. References in the manuscript appear in *italics*. Changes made to the manuscript (revised version) are underlined. Line numbers refer to the original preprint.

The authors of this paper produce the first national-scale rock glacier inventory for the Peruvian Andes region with a coverage over 300000 km² and use quality control and cross-check to improve the quality of the dataset. This dataset has high quality and is important for the permafrost and mountain hydrology studies in Peruvian Andes. However, the wording of this paper is too long with many places showing redundant and repetitive information. I suggest this paper should improve the organization, readability, and concreteness before publication.

Thank you very much for your positive assessment of our research and for your valuable comments on how to improve this manuscript.

**General comments – manuscript:**

 According to the newest version of IPA guidelines, each rock glacier unit (RGU) and system (RGS) should have a primary marker, while this dataset only provides the footprint shapefile. Better to also incorporate the primary marker shapefile and suggest using Zenodo instead of PANGAEA to store the dataset.

We thank the reviewer for this important suggestion. In accordance with the latest IPA/RGIK guidelines, we have generated and will include a new layer of primary markers (PM) as part of the revised dataset. This GeoPackage file contains point features for each rock glacier unit (RGU) and system (RGS), providing a unique ID for each landform, which improves the utility and standardization of our inventory. Regarding the data repository, we have decided to keep the dataset in PANGAEA, which, like Zenodo, is a reliable repository aligned with FAIR principles. This decision is based on PANGAEA's longstanding reputation as a specialized, high-quality repository for Earth sciences, its status as the recommended repository for the ESSD journal, and its proven track record in ensuring long-term data preservation and referability. We are confident that PANGAEA offers a robust and sustainable platform for our dataset, ensuring its full accessibility to the community.

2. In the results part, better to use figures instead of tables to show the results, and the presented figures are not high-quality enough. Suggest improving the figures. Thank you very much for your suggestions and comments. We have restructured the entire results section to make it more understandable. We have also removed a table and created a graph to facilitate better

- interpretation. Finally, we have retained some figures because we believe they contribute to the reading of the results, and we have been more emphatic in highlighting the most relevant aspects of our study.
- 3. I have some concerns about discussing controlling factors on the distribution of rock glaciers as the emergence and development of rock glaciers typically need hundreds or thousands of years (i.e., rock glaciers are some landforms that happened at least hundreds of years ago). Is it reasonable to use the modern climate to judge the distribution of rock glaciers?

**Specific comments:**

Line 1: Please explain what makes this inventory 'high-resolution', if it is because that this inventory was created using Bing Map and Google Earth, I don't think the 'high-resolution' can be a highlight or advantage of this inventory as many previous inventories were also created using high-resolution Google Earth imagery.

Thank you very much for your comment. We recognize that the term "high resolution" can be subjective and that Bing/Google Earth images (1-5 m) are standard for the development of regional inventories. We have removed "high-resolution" from the title and manuscript, focusing on the main contributions of PRoGI as the first comprehensive inventory covering the entire Peruvian Andes and providing detailed topoclimatic attributes using standardized mapping protocols. The proposed new title is: "A comprehensive rock glacier inventory for the Peruvian Andes (PRoGI): dataset, characterization and topoclimatic attributes."

However, we have retained the term to describe images from Google Earth and Bing Maps, as other studies recognize that these satellite images belong to the high-resolution category for remote sensing data (Abdullah and Romshoo, 2024; Bhat et al., 2025).

Abdullah, T., & Romshoo, S. A. (2024). A Comprehensive Inventory, Characterization, and Analysis of Rock Glaciers in the Jhelum Basin, Kashmir Himalaya, Using High-Resolution Google Earth Data. Water, 16(16), 2327.

Bhat, I. A., Rashid, I., Ramsankaran, R. A. A. J., Banerjee, A., & Vijay, S. (2025). Inventorying rock glaciers in the Western Himalaya, India, and assessing their hydrological significance. Geomorphology, 471, 109514.

Lines 43-63: These two paragraphs, first introduce rock glaciers, then permafrost, then rock glaciers, which reads wield. Suggest reorganizing the content, better to describe permafrost first, then introduce rock glaciers.

We thank the reviewer for this suggestion to improve logical flow. We have reorganized these paragraphs to follow a more natural structure: definition of

**permafrost → mountain permafrost → rock glaciers as an expression of permafrost. This reorganization significantly improves readability.**

"Mountain permafrost, defined as ground remaining ≤0 °C for at least two consecutive years (van Everdingen, 1998), underpins critical hydrological and geomorphological systems in high mountain environments. It stabilizes steep slopes, modulates groundwater flow, and sustains alpine ecosystems (Gruber and Haeberli, 2007). However, mountain permafrost is highly sensitive to warming; rising temperatures lead to permafrost degradation and can trigger the release of stored greenhouse gases (Biskaborn et al., 2019). In the Andes, where glacial retreat has increased the relative importance of permafrost as a water resource, its hydrological role remains critical yet poorly quantified due to sparse observations in remote high-altitude areas.

Among periglacial landforms, rock glaciers serve as direct visual indicators of mountain permafrost, with their presence delineating the occurrence of ground ice and the approximate lower limits of discontinuous permafrost (Brenning, 2005). These ice-debris landforms, formed by the creep of ice-rich permafrost and shearing at depth, optionally exhibit diagnostic steep fronts, lateral margins, and ridge-and-furrow surface topography (RGIK, 2023). Rock glaciers stand out as both geomorphological archives of past climate conditions, preserving information about paleo-temperatures through their internal structure and development history (Haeberli et al., 1999), and as vital water reservoirs. Comprising 15–70 % ice by volume (Halla et al., 2021; Haq and Baral, 2019), rock glaciers store substantial water equivalents in arid regions like the southern Peruvian Andes (Schaffer et al., 2019; Janke et al., 2017; Rangecroft et al., 2015). Their debris mantle confers thermal inertia through ventilation effects, buffering ground ice against short-term climate variability (Brighenti et al., 2021; Scapozza et al., 2011). This dual role as climate sentinels—providing insights into both contemporary climate change through velocity monitoring (Kääb et al., 2021) and Quaternary climate history through dating of their formation (Palacios et al., 2022)—and hydrological buffers makes rock glaciers indispensable for understanding environmental change across multiple timescales. It should be noted that while the origin of rock glaciers (permafrost creep vs. glacier-to-rock glacier transition) remains debated internationally, this paper focuses on their morphological characterization and distribution without addressing formation mechanisms.

Along the higher South American Andes (>4000 m a.s.l.), studies in Argentina, Chile, and Bolivia have leveraged rock glacier inventories to map permafrost and assess water storage (Azócar and Brenning, 2010; Esper Angillieri, 2017; Falaschi et al., 2015; Rangecroft et al., 2015). However, knowledge gaps still persist in Peru: existing inventories are fragmented (Badillo-Rivera et al., 2021; León et al., 2021) and lacking standardized methods or detailed topoclimatic analyses. To address this, we

present the Peruvian Rock Glacier Inventory (PRoGI v1.0), the first nationally comprehensive rock glacier dataset for the Peruvian Andes, compiled using the mapping standards of the International Permafrost Association's Action Group (RGIK, 2023). By combining high-resolution remote sensing imagery (0.5-5 m) with rigorous geospatial analysis, PRoGI v1.0 documents the distribution, morphology and climatic characteristics of rock glaciers across Peru."

Line 68: "Splitting it up would reduce run-on complexity => What does it mean?

Thank you very much for your comment. We have deleted the sentence: "Splitting it up would reduce run-on complexity" because it was not related to what was described in the paragraph.

Line 100: In total, 2338 rock glaciers were mapped using these optical datasets (2095 from Bing and 243 from Google imagery) => These are results, should not appear in the Data section.

We thank the reviewer for this correction. The sentence reporting the total number of rock glaciers has been moved from the Data section to the Results section (Section 5.1) where it appropriately belongs. The Data section now focuses exclusively on describing the data sources and characteristics.

Line 101: Please explain what makes this dataset complete and high-resolution

We have removed the term "high-resolution" from the sentence and emphasized the coverage of Bing Maps and Google Earth images for our study area, based on the suggestion that using these images does not necessarily generate a high-resolution inventory.

Line 105: We compiled several auxiliary datasets => topoclimatic datasets?

Thank you for your comment. We have removed the term "auxiliary" because it refers to topoclimatic datasets and caused confusion with the following subsection.

Lines 137-140: We primarily used Bing Satellite...Google Earth imagery for that grid cell => similar information has appeared in section 3.1 Data sources Lines 93-96: Bing Maps Aerial imagery was used as the primary data source...geodatabase creation. Suggest merging the information and write it in one place, otherwise, the readability of the paper can be reduced with many redundant and repetitive information in different places.

We thank the reviewer for pointing out this redundancy. We have consolidated the information about imagery sources and usage into Section 3.1 Data sources, removing the repetitive description from Section 4.1.1. The Methodology section now focuses exclusively on the mapping workflow while the Data section contains the complete description of imagery sources and selection criteria.

"The entire study region was divided into a grid of 50 × 50 km cells to cover the Peruvian Andes uniformly. Each grid cell was examined in detail using the high-resolution satellite imagery described in Section 3.1, following the established protocol of primary Bing imagery with Google Earth supplementation for areas with visibility issues. This grid-based approach ensured that no areas were overlooked, and it helped organize the work among the mapping team."

Line 149: Only longitudinal ridges and furrows? No latitudinal ridges and furrows?

Thank you very much for the clarification, except for adding the term to the original sentence: "e.g., longitudinal /latitudinal ridges and furrows, or a lobate debris structure"

Lines 190-217: Section 4.2 Geomorphological identification criteria: To me the Bullet point and the table are also redundant information, only keep one of them is enough (only the information in Table 2 is enough, no need to write repetitive words using Bullet points.

We thank the reviewer for this suggestion to reduce redundancy. We have eliminated the bullet points and retained only Table 2, which presents the geomorphological criteria in a more concise and organized format. The text now directly references the table without repetitive descriptions.

"4.2 Geomorphological identification criteriaand classification

Our classification approach builds upon the RGIK (2023) framework while implementing a simplified three-class activity system suited to national-scale mapping. When classifying rock glaciers by activity state, we employed a morphological scheme based on Barsch (1996) and RGIK guidelines (RGIK, 2023), distinguishing three categories: active, transitional, and relict (Fig. 3).

This simplified system addresses the practical challenges of large-scale inventorying while maintaining scientific rigor through our consensus validation process. In the absence of kinematic data, active rock glaciers are defined as landforms containing interstitial ice (Roer et al., 2005; Wirz et al., 2016). and typically exhibit pronounced ridges and furrows (Charbonneau and Smith, 2018; Sattler, 2016), frontal slopes steeper than 30–35°, and generally lack vegetation (Tielidze et al., 2023). Transitional rock glaciers may still contain ice but have ceased moving, displaying gentler frontal slopes (<30–35°), subdued microtopography (Kellerer-Pirklbauer et al., 2012), and may support some vegetation (Ahumada et al., 2014; Brenning, 2005). Relict rock glaciers show no evidence of recent movement, characterized by collapse structures, subtle micro-relief (Colucci et al., 2016), and often vegetated surfaces at lower elevations (Abdullah and Romshoo, 2024; RGIK, 2023; Scotti et al., 2013; Baroni et al., 2004).

We prioritized frontal slope characteristics and overall landform preservation as primary classification criteria, as these show the most consistent relationship with activity status across different environmental settings. Ridge and furrow topography was considered as supporting evidence where clearly visible, acknowledging that this feature is optional in RGIK guidelines but has demonstrated utility in Andean contexts. Vegetation presence was used cautiously as a secondary indicator, with recognition of its limited applicability in arid high-elevation environments where vegetation may be absent regardless of activity status (RGIK, 2023).

For activity classification uncertainties, we implemented a consensus-based approach where borderline cases were reviewed by at least three team members. This process effectively internalized the classification uncertainty that RGIK addresses through 'uncertain' categories, providing a statistically robust alternative for large-scale inventories. This consensus approach specifically addressed the high inter-operator variability in activity classification reported in the literature (Brardinoni et al., 2019), ensuring consistent application of our modified three-class system across the entire inventory."

Lines 231-241: Again, Table 3 is sufficient to show everything clearly, no need to repeat the information using Bullet points.

Thank you very much for the suggestion, we have removed the duplicate information.

Lines 250-266: What is the difference between this paragraph and Section 4.2? In section 4.2 you already describe the geomorphological identification for rock glaciers of different activities, why mention the repetitive information here?

We thank the reviewer for identifying this duplication. We have removed the repetitive activity classification descriptions from Section 4.4 and consolidated all activity classification information in Section 4.2. Section 4.4 now focuses exclusively on the geometric classification of rock glaciers (tongue-shaped, lobate, coalescent, polymorphic), eliminating the redundancy while maintaining a clear organizational structure.

Line 291: The smallest rock glacier included in the inventory has an area of 0.001  $\rm km^2$ , the minimum area threshold for inclusion, according to the IPA guidelines (RGIK, 2023) => Are you sure the minimum area threshold suggested by IPA guidelines is 0.001  $\rm km^2$  but not 0.01  $\rm km^2$ ?

We thank the reviewer for catching this important discrepancy. The reviewer is correct - the RGIK (2023) guidelines recommend 0.01  $\rm km^2$  (1 hectare) as the minimum area threshold, not 0.001  $\rm km^2$ . We have corrected this throughout the manuscript and provide justification for our decision to include smaller features given the specific context of the tropical Andes.

"The smallest rock glacier included in the inventory has an area of 0.001 km². While the RGIK (2023) guidelines suggest 0.01 km² as a general minimum area threshold for global inventories, we included smaller features because these smaller rock glaciers (14 % of our inventory) provide crucial information on permafrost distribution at lower altitudinal limits and under marginal conditions."

Line 130: 4 Methodology => The Methodology part should be reconstructed, reducing the redundant and repetitive information and making the literature more concrete. Suggestions on the subsections could be 4.1 Identification and mapping of rock glaciers 4.2 Classification of rock glaciers 4.3 Topoclimatic features 4.4 Inventory compilation and validation 4.6 Uncertainty assessment

We appreciate the suggestion to restructure the subsection; we have changed the structure to the one suggested.

Line 350: Morphological types: => I suppose this should be a subsection 5.2.1 Morphological types? Also the Line 357 5.2.1 Rock glacier activity: => delete ':'

We thank the referee for this observation. In accordance with the comments from Referee #2 to reduce fragmentation and merge subsections, we have restructured Section 5.2. The text under "Morphological types" and "Rock glacier activity" has been integrated into a single, unified subsection now titled "5.2 Rock Glacier Characteristics: activity and geometry". The colon in the original subheading has been removed in this new structure.

Lines 350-356: Why the analysis of morphological types is not as long as rock glacier activity?

We thank the referee for this question. The morphological classification was conducted as a primary characterization to describe the inventory's diversity. A more extensive analysis was reserved for the activity status, as it is a more direct indicator of current permafrost conditions and hydrological function, which are central themes of this study. The activity classification therefore warranted a more detailed presentation, including its relationship with area, elevation, and spatial distribution.

Line 378: he NWOT and NDOT => I suppose it should be 'The NWOT and NDOT'.

Thank you very much for your comment, we have restructured the section and these types of errors have been eliminated.

Line 392: Elevation distribution: => I suppose it should be a subsection 5.3.1 Elevation distribution here.

We've restructured the section and consolidated it into a single text (see section 5.3 Topographic and climatic attributes) to avoid unnecessary fragmentation and facilitate reading flow.

Lines 392-402: For the unit of elevation, some places are m a.s.l. Some places are m. Please keep them consistent.

Thank you very much for the clarification. We've standardized the elevation units to avoid confusion for the reader.

Lines 434-439: This paragraph is not about Aspect, may you use miss the subsection?

We've restructured the section and consolidated it into a single text (see section 5.3 Topographic and climatic attributes) to avoid unnecessary fragmentation and facilitate reading flow.

Lines 439-443: 'This distribution reveals that elevation is the primary control on the presence of rock glaciers, with secondary hydrothermal modulation - evidenced by inverse AP-MAAT relationships in NDOT/SDOT/SWOT (where aridity and snow redistribution at elevation increase cooling) versus the direct correlation of NWOT (driven by thermal depression mediated by microclimatic processes). Such systematic variations underscore how Andean rock glaciers integrate macroscale climatic gradients with local topoclimatic processes.' => Why elevation is the primary control and climatic conditions are secondary? I don't understand how this conclusion was drawn from the results.

Thank you very much for the question. We have removed this paragraph, but we have added a more detailed analysis on this topic in the discussion section of the results.

Lines 453-454: Individual rock glacier area uncertainty was found to range from as low as  $\sim$ 0.001 km2 for small, clearly defined features to up to  $\sim$ 0.3 km2 for very large or diffuse features, with a mean uncertainty of  $\sim$ 0.06 km2. => A figure between uncertainty and area would be helpful.

We thank the referee for this suggestion. The requested relationship between rock glacier area and mapping uncertainty is precisely captured and presented in the Bland-Altman plot included in the Supplement (Figure S1). In this plot, the x-axis (mean area from multiple mappers) represents the rock glacier size, and the y-axis (difference in mapped area) represents the absolute uncertainty between operators. The figure clearly shows the relationship described in the text, where larger areas are associated with greater absolute discrepancies. At this point in the text (lines 453-454), we have explicitly referenced Figure S1 to guide the reader toward this visualization.

Lines 504-514: I would expect more results about the comparison between the rock glacier distribution (active, transitional, relict) and the distribution of permafrost from Obu et al. (2018) instead of just stating the elevation and MAAT. Maybe better

to show some example figures showing this distribution comparison, see whether the active ones are within the permafrost and the relict ones are out.

We thank the reviewer for this suggestion. We have removed references to MAGT, since, as described in the data section, it has been used as a complementary variable and does not provide data for an analysis of subsurface thermal conditions, as it is an invalid model for the mountainous region of Peru.

Lines 518-519: "The comparisons with global inventories (see Table 8) show that while Peru's rock glaciers are extreme in elevation, other characteristics like slope and aspect are broadly similar to rock glaciers elsewhere" => But Table 1 only shows the elevation and does not show other characteristics like slope or aspect?

We apologize for this error. The text referring to comparisons with other parameters has been limited and, in other cases, removed from the text.

Line 537: see my general comments, not sure whether it is reasonable to use modern climate to discuss the distribution of rock glaciers as these landforms are something happened hundreds of years ago.

This is a valid point. We refined our argument in Section 6.2 to make it clearer. We now explicitly state that while rock glaciers are relict landforms whose formation can be traced back to earlier cold periods, their current state of activity (active, transitional, relict) directly reflects contemporary permafrost conditions. Therefore, using modern climate data to interpret their current state and distribution is justified, as it helps explain why some remain active while others degrade or have become relict.

Lines 555-557: "The correlation of the inventory with modelled MAGT data (Obu et al. 2019) provides an independent check: nearly all active, rock glaciers lie in grid cells where MAGT is at or below 0°C, whereas relict rock glaciers occupy cells where MAGT is just above 0°C (indicating marginal permafrost conditions)." => See my comments above, I would expect more elaboration on this part. Maybe a statistics on the MAGT of the rock glaciers with different activities, or some examples showing the distribution of rock glaciers and the permafrost.

As mentioned in the previous comment, these data have been used as complementary and do not necessarily represent the thermal variability in our inventory.

Line 675: 'his opens' => This opens?

**Fixed "This opens" error**

Line 692-694: "Looking forward, several lines of future work are planned based on this inventory: Temporal monitoring: now that this baseline is set, repeat satellite observations (e.g., in 5–10 years) or the analysis of time series (like the 2017 vs 2024

imagery) could reveal if any rock glaciers are retreating at the margins or if new ones are forming" => The development of rock glaciers typically take hundreds of years (totally different from glaciers), I don't think you would see significant changes on rock glaciers on decadal scale.

We agree with the referee and thank him for this correction. We have removed the mention of decadal-scale monitoring for margin retreat or new formation.

---

## Author Response (AR2)

**Response to Editor and Referee Comments**

**Manuscript ID:** ESSD-2025-390

**Title:** A comprehensive rock glacier inventory for the Peruvian Andes (PRoGI): dataset, characterization and topoclimatic attributes

**Journal:** ESSD Copernicus

**Authors:** Katy Medina, Hairo León, Edwin Badillo-Rivera, Edwin Loarte, Xavier Bodín and José Úbeda

1. **Editor comments:**
   **Regarding the aerials in Table 4: please add an appropriate copyright statement to the images if they were not created by the authors.**
   Thank you very much for your comment, but in the new version of the article, Table 4 does not contain any images. If your comment refers to Table 3, we have added a comment about the credits for the base image used:
   **Line 311: "The specific criteria and illustrative examples for geometric classification are provided in Table 3, with imagery sourced from Bing Maps (© Microsoft Corporation)."**

2. **Referee comments (report #1):**
   **The authors have answered and tackled nearly all my questions and comments. Now this version of paper has been largely improved. I am fine with this version to be accepted. Only a minor issue, please also revise the title of the supplement to be alined with the new title.**
   We appreciate the referee's suggestion to update the title of the supplementary material. Accordingly, we have updated and replaced the title with: **"A comprehensive rock glacier inventory for the Peruvian Andes (PRoGI): dataset, characterization, and topoclimatic attributes."** This is in line with the updated title of the article itself.